# Aligning Target-Aware Molecule Diffusion Models with Exact Energy Optimization

**Siyi Gu**[*1], **Minkai Xu**[*1†], **Alexander Powers**[1], **Weili Nie**[2], **Tomas Geffner**[2]
**Karsten Kreis**[2], **Jure Leskovec**[1], **Arash Vahdat**[2], **Stefano Ermon**[1]
[1] Stanford Univeristy [2] NVIDIA
{sgu33,minkai,jure,ermon}@cs.stanford.edu   lxpowers@stanford.edu
{wnie,tgeffner,kkreis,avahdat}@nvidia.com

## Abstract

Generating ligand molecules for specific protein targets, known as structure-based drug design, is a fundamental problem in therapeutics development and biological discovery. Recently, target-aware generative models, especially diffusion models, have shown great promise in modeling protein-ligand interactions and generating candidate drugs. However, existing models primarily focus on learning the chemical distribution of all drug candidates, which lacks effective steerability on the chemical quality of model generations. In this paper, we propose a novel and general alignment framework to align pretrained target diffusion models with preferred functional properties, named ALIDIFF. ALIDIFF shifts the target-conditioned chemical distribution towards regions with higher binding affinity and structural rationality, specified by user-defined reward functions, via the preference optimization approach. To avoid the overfitting problem in common preference optimization objectives, we further develop an improved Exact Energy Preference Optimization method to yield an exact and efficient alignment of the diffusion models, and provide the closed-form expression for the converged distribution. Empirical studies on the CrossDocked2020 benchmark show that ALIDIFF can generate molecules with state-of-the-art binding energies with up to -7.07 Avg. Vina Score, while maintaining strong molecular properties. Code is available at https://github.com/MinkaiXu/AliDiff.

## 1 Introduction

Generating ligand molecules with desirable properties and high affinity to specific protein targets, known as structure-based drug design (SBDD), is a fundamental problem in therapeutic design and biological discovery. It necessitates methods that can produce realistic and diverse drug-like molecules with stable 3D structures and high binding affinities. In the past few years, numerous deep generative models have been proposed to generate molecules in SMILES string representation [Kusner et al., 2017, Segler et al., 2018] or graph representations [Jin et al., 2018, Shi et al., 2020, Guan et al., 2023]. Although these models have shown promise in generating plausible drug-like molecules, they lack sufficient modeling of the 3D protein-ligand interaction with proteins and therefore can hardly be adopted in target-aware molecule generation. As a result, generating ligands conditioned on protein targets remains an open research problem.

Recently, with rapid progress in structural biology and the increasing scale of structural data [Francoeur et al., 2020, Jumper et al., 2021], numerous target-aware generative models have been proposed to directly generate molecules within the protein targets in 3D. Initial work proposed to sequentially

---

*Equal contribution; junior author listed earlier. †Correspondence to: Minkai Xu <minkai@cs.stanford.edu>.

38th Conference on Neural Information Processing Systems (NeurIPS 2024).

place atoms within the target via autoregressive models [Luo et al., 2021, Liu et al., 2022, Peng et al., 2022], while later work learns diffusion models to jointly design the whole ligand with state-of-the-art results [Guan et al., 2023, Lin et al., 2022, Schneuing et al., 2023, Huang et al., 2023, Guan et al., 2024]. Following the biological principle to model the protein-ligand complex interactions, these methods have shown great promise in generating realistic drugs that can bind toward given targets. However, all existing models solely focus on learning the chemical distribution of candidate molecules and treat all training samples equally, while in practice, only the ligand molecules with strong binding affinity and high synthesizability are preferred for real-world therapeutic development. As a result, existing learned models generally lack sufficient steerability regarding the relative quality of model generations and cannot generate faithful samples with the desirable properties.

To bridge the gap between existing SBDD models and the necessity for designing ligands with favorable properties, in this paper, we introduce a novel and comprehensive alignment framework to align pretrained target-aware diffusion models with preferred functional properties, named ALIDIFF. ALIDIFF adjusts the target-conditioned chemical distribution toward regions characterized by lower binding energy and structural rationality, as specified by a user-defined reward function, using a preference optimization approach. To this end, we derive a unified variational lower bound to align the likelihoods of both discrete chemical type and continuous 3D coordinate features. We further analyze the winning data overfitting problem commonly associated with preference optimization objectives, and introduce an improved Exact Energy Preference Optimization ($\text{E}^2\text{PO}$) method. $\text{E}^2\text{PO}$

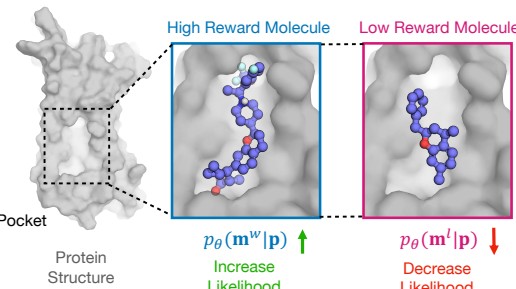

Figure 1: High-level illustration of ALIDIFF. For a protein target, we can have multiple candidate ligands and rank the preference by certain reward functions, *e.g.*, binding energy. We align the target-aware molecule diffusion model with these preferences by adjusting the conditional likelihoods.

analytically ensures a precise and efficient alignment of diffusion models, and we provide a closed-form expression for the converged distribution. Our key contributions can be summarized as follows:

- We address the challenge of designing favorable target-aware molecules from the perspective of aligning molecule generative models with desirable properties. We introduce the energy preference optimization framework and derive variational lower bounds to align diffusion models for generating molecules with high binding affinity to binding targets.

- We analyze the overfitting issue in the preference optimization objective, and propose an improved exact energy optimization method to yield an exact alignment towards target distribution shifted by reward functions.

- We conduct comprehensive comparisons and ablation studies on the CrossDocked2020 [Francoeur et al., 2020] benchmark to justify the effectiveness of ALIDIFF. Empirical results demonstrate that ALIDIFF can generate molecules with state-of-the-art binding energies with up to -7.07 Avg. Vina Score, while maintaining strong molecular properties.

## 2 Related Work

**Structure-Based Drug Design.** With increasing amount of structural data becoming accessible, generative models have attracted growing attention for structure-based molecule generation. Early research [Skalic et al., 2019] proposes to generate SMILES representations from protein contexts by sequence generative models. Inspired by the progress in 3D and geometric modeling, many works proposed to solve the problem directly in 3D space. For instance, Ragoza et al. [2022] voxelizes molecules within atomic density grids and generates them through a Variational Autoencoder framework. Luo et al. [2021], Peng et al. [2022], Liu et al. [2022], Powers et al. [2023] developed autoregressive models to generate molecules by sequentially placing atoms or chemical groups within the target. Following the autogressive backbone, FLAG[Zhang et al., 2023] and DrugGPS [Zhang and Liu, 2023] take advantage of chemical priors of molecular fragments to generate ligand molecules piece by piece, leading to more realistic substructures. More recently, diffusion models achieved exceptional results in synthesizing high-quality images and texts, which have also been successfully

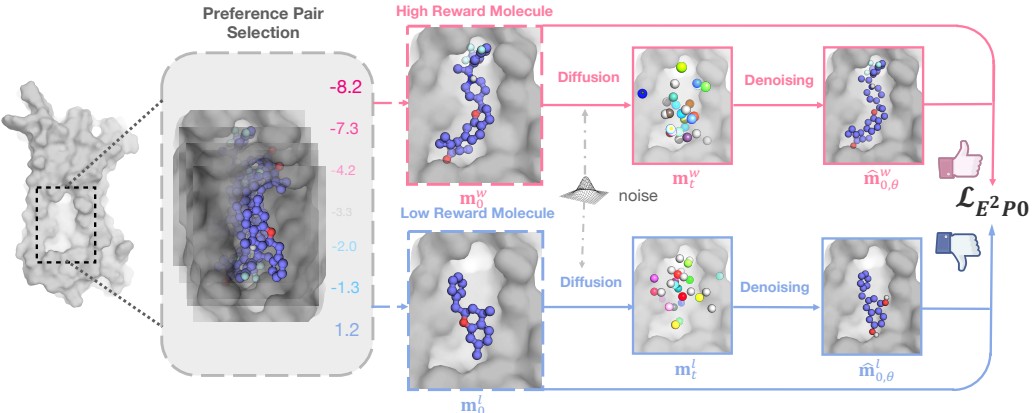

Figure 2: Overview of ALIDIFF. This workflow can be summarized as 1) For each protein target (pocket) $\mathbf{p}$ in the training set, we retrieve two candidate ligands $\mathbf{m}$; 2) Label the two ligands as wining sample $\mathbf{m}^w$ and losing sample $\mathbf{m}^l$ by desirable properties, *e.g.*, binding energies; 3) Calculate the preference optimization objective Equation (12) and update the molecule diffusion model $p_\theta$.

used for ligand molecule generation [Guan et al., 2023, Lin et al., 2022, Schneuing et al., 2023, Huang et al., 2023, Guan et al., 2024]. These models generate molecules by progressively denoising atom types and coordinates while maintaining physical symmetries with SE(3)-equivariant neural networks. While the existing works focus on designing molecules using various deep generative models, they often struggle with generating molecules that exhibit different desirable properties, *e.g.*, strong binding affinity, high synthesizability, and low toxicity. Real-word drug discovery projects almost always seek to optimize or constrain these properties [D Segall, 2012, Bickerton et al., 2012]. In this work, we aim to address the challenge with a novel and general preference optimization framework.

**Reinforcement learning from human feedback (RLHF).** Recently, significant efforts have been devoted to aligning generative models with human preferences. The use of reinforcement learning to incorporate feedback from humans and AI into finetuning large language models is exemplified by Reinforcement Learning from Human Feedback (RLHF) [Ziegler et al., 2020, Ouyang et al., 2022]. Research works have incorporated human feedback to improve performance across various domains, such as machine translation [Nguyen et al., 2017], summarization [Stiennon et al., 2020], and also diffusion models [Uehara et al., 2024b,a]. Notably, Rafailov et al. [2023] designed a new preference paradigm that enables training language models to satisfy human preferences directly without reinforcement learning. This algorithm was later applied to diffusion models for text-to-image generation tasks [Wallace et al., 2023]. Concurrent work [Zhou et al., 2024] attempts to apply DPO for designing antibodies with rationality and functionality. To the best of our knowledge, we are the first alignment approach for target-aware ligand design, where the conditional distribution is shifted toward desirable properties.

## 3 Method

In this section we present ALIDIFF, a general framework for aligning target-aware diffusion models with various molecular functionalities. We first provide an overview of the target-aware ligand diffusion model and our Reinforcement Learning from Feedback formulation (section 3.1). Next, we introduce the energy optimization approach for aligning the diffusion model and analyze the potential limitations of the framework (section 3.2). We then further introduce an exact energy optimization method from a distribution matching perspective to align the generative model efficiently and exactly (section 3.3). A visualization of the framework is shown in Figure 2.

### 3.1 Overview

**Notation.** We focus on aligning molecule generative models for structure-based drug design, which can be abstracted as generating molecules that can bind to a given protein target. Following the

convention in the related literature [Luo et al., 2021, Guan et al., 2023], the molecule and target protein are represented as $\mathcal{M} = \{(\mathbf{x}_M^{(i)}, \mathbf{v}_M^{(i)})\}_{i=1}^{N_M}$ and $\mathcal{P} = \{(\mathbf{x}_P^{(i)}, \mathbf{v}_P^{(i)})\}_{i=1}^{N_P}$, respectively, where $N_M$ and $N_P$ denote the number of atoms of the molecule $\mathcal{M}$ and the protein $\mathcal{P}$. $\mathbf{x} \in \mathbb{R}^3$ and $\mathbf{v} \in \mathbb{R}^K$ denote the atomic 3D position and chemical type, respectively, with $K$ being the dimension of atom types. For brevity, we denote the molecule as a matrix $\mathbf{m} = [\mathbf{x}_M, \mathbf{v}_M]$ where $\mathbf{x}_M \in \mathbb{R}^{N_M \times 3}$ and $\mathbf{v}_M \in \mathbb{R}^{N_M \times K}$, and denote the protein as a matrix $\mathbf{p} = [\mathbf{x}_P, \mathbf{v}_P]$ where $\mathbf{x}_P \in \mathbb{R}^{N_P \times 3}$ and $\mathbf{v}_P \in \mathbb{R}^{N_P \times K}$. The task can then be formulated as modeling the conditional distribution $p(\mathbf{m}|\mathbf{p})$.

**Preliminaries.** Diffusion Models have been previously used to model the joint distribution of atomic types and positions [Guan et al., 2023, Schneuing et al., 2023, Lin et al., 2022]. This approach consists of a forward diffusion process and a reverse generative (denoising) process. Both processes are only defined on the ligand molecules $\mathbf{m}$, with fixed proteins $\mathbf{p}$. In the forward process, small Gaussian and categorical noises are gradually injected on atomic coordinates $\mathbf{x}$ and types $\mathbf{v}$ as follows:

$$q(\mathbf{m}_t|\mathbf{m}_{t-1}, \mathbf{p}) = \mathcal{N}(\mathbf{x}_t; \sqrt{1-\beta_t}\mathbf{x}_{t-1}, \beta_t \boldsymbol{I}) \cdot \mathcal{C}(\mathbf{v}_t; (1-\beta_t)\mathbf{v}_{t-1} + \beta_t/K), \tag{1}$$

where $\mathcal{N}$ and $\mathcal{C}$ stand for the Gaussian and categorical distribution respectively, and $\beta_t$ corresponds to a (fixed or learnable) variance schedule. Note that, in certain recent work $q$ process can be learnable with dependence on the conditioning $\mathbf{p}$ [Huang et al., 2023]. We omit the subscript $M$ for the ligand molecule without ambiguity here and denote the atom positions and types at time step $t$ as $\mathbf{x}_t$ and $\mathbf{v}_t$. Using Bayes theorem, the posterior conditioned on $\mathbf{m}_0$ can be computed in closed form:

$$q(\mathbf{m}_{t-1}|\mathbf{m}_t, \mathbf{m}_0, \mathbf{p}) = \mathcal{N}(\mathbf{x}_{t-1}; \tilde{\boldsymbol{\mu}}(\mathbf{x}_t, \mathbf{x}_0), \tilde{\beta}_t \boldsymbol{I}) \cdot \mathcal{C}(\mathbf{v}_{t-1}; \tilde{\boldsymbol{c}}(\mathbf{v}_t, \mathbf{v}_0)), \tag{2}$$

where $\tilde{\boldsymbol{\mu}}(\mathbf{x}_t, \mathbf{x}_0) = \frac{\sqrt{\bar{\alpha}_{t-1}}\beta_t}{1-\bar{\alpha}_t}\mathbf{x}_0 + \frac{\sqrt{\alpha_t}(1-\bar{\alpha}_{t-1})}{1-\bar{\alpha}_t}\mathbf{x}_t$, $\tilde{\beta}_t = \frac{1-\bar{\alpha}_{t-1}}{1-\bar{\alpha}_t}\beta_t$, $\alpha_t = 1-\beta_t$, $\bar{\alpha}_t = \prod_{s=1}^t \alpha_s$, $\tilde{\boldsymbol{c}}(\mathbf{v}_t, \mathbf{v}_0) = \frac{\boldsymbol{c}^*}{\sum_{k=1}^K c_k^*}$, and $\boldsymbol{c}^*(\mathbf{v}_t, \mathbf{v}_0) = [\alpha_t \mathbf{v}_t + (1-\alpha_t)/K] \odot [\bar{\alpha}_{t-1}\mathbf{v}_0 + (1-\bar{\alpha}_{t-1})/K]$ [Ho et al., 2020, Austin et al., 2021]. At timestep $T$, $q$ converges to the prior with Gaussians on coordinates and uniforms on atom types. The reverse process, also known as the generative process, learns a neural network parameterized by $\theta$ to recover data by iterative denoising. The denoising step can be approximated with predicted Gaussians $\boldsymbol{\mu}_\theta$ and categorical distributions $\boldsymbol{c}_\theta$ as follows:

$$\begin{aligned} p_\theta(\mathbf{m}_{t-1}|\mathbf{m}_t, \mathbf{p}) &= \mathcal{N}(\mathbf{x}_{t-1}; \boldsymbol{\mu}_\theta([\mathbf{x}_t, \mathbf{v}_t], t, \mathbf{p}), \tilde{\beta}_t \boldsymbol{I}) \cdot \mathcal{C}(\mathbf{v}_{t-1}; \boldsymbol{c}_\theta([\mathbf{x}_t, \mathbf{v}_t], t, \mathbf{p})) \\ &= \mathcal{N}(\mathbf{x}_{t-1}; \tilde{\boldsymbol{\mu}}(\mathbf{x}_t, \hat{\mathbf{x}}_0), \tilde{\beta}_t \boldsymbol{I}) \cdot \mathcal{C}(\mathbf{v}_{t-1}; \tilde{\boldsymbol{c}}(\mathbf{v}_t, \hat{\mathbf{v}}_0)), \end{aligned} \tag{3}$$

where $[\hat{\mathbf{x}}_0, \hat{\mathbf{v}}_0] = \epsilon_\theta([\mathbf{x}_t, \mathbf{v}_t], t, \mathbf{p})$ are predictions from a denoising network $\epsilon_\theta$. Importantly, the denoising network here is specifically parameterized by equivariant neural networks, resulting in an SE(3)-invariant likelihood $p_\theta(\mathbf{m}|\mathbf{p})$ on the protein-ligand complex [Xu et al., 2022].

**Overview.** As ligand molecules with desirable properties, *e.g.*, high binding affinity and synthesizability, are required for real-world therapeutic development, we aim to align the ligand diffusion model with these preferences. Such preferences can be defined as a reward model $r(\cdot) : \mathcal{M} \times \mathcal{P} \to \mathbb{R}$ calculated from various cheminformatics software, *e.g.*, binding affinity, drug-likeness, synthesizability, or their combinations. We fine-tune and align the pre-trained diffusion model with the reinforcement learning framework. Specifically, given a dataset $\mathcal{D}$ containing given protein targets, inspired by RLHF [Ouyang et al., 2022], this fine-tuning is achieved by maximizing the reward:

$$\max_{p_{\boldsymbol{\theta}}} \mathbb{E}_{\mathbf{p} \sim \mathcal{D}, \mathbf{m} \sim p_{\boldsymbol{\theta}}}[r(\mathbf{m}, \mathbf{p})] - \beta \mathbb{D}_{\text{KL}}(p_{\boldsymbol{\theta}}(\mathbf{m}|\mathbf{p}) \| p_{\text{ref}}(\mathbf{m}|\mathbf{p})), \tag{4}$$

where $p_{\boldsymbol{\theta}}$ and $p_{\text{ref}}$ are the distributions induced by the fine-tuned and pre-trained models, respectively. In this work, $p_{\boldsymbol{\theta}}$ and $p_{\text{ref}}$ are the fine-tuned and pre-trained molecule diffusion models, as introduced above. $\beta$ is a hyperparameter controlling the KL divergence regularization. Note that, here the reward is a known black-box function, unlike typical RLHF where it is unknown and has to be estimated from preferences. In the following section, we elaborate on how the alignment objective is rewritten with diffusion forward and reverse processes defined on atomic types and coordinates.

## 3.2 Energy Preference Optimization

Though the reward function is known, evaluating reward values such as binding affinity is computationally expensive and we instead resort to aligning with a labeled offline dataset. We start with a dataset $\mathcal{D} = \{(\mathbf{p}, \mathbf{m}^w, \mathbf{m}^l)\}$ where $\mathbf{p}$ denotes the protein condition and $\mathbf{m}^w \succ \mathbf{m}^l$ is a pair of winning and losing ligands with respect to certain specified energy, *e.g.*, binding energy.

The optimal solution to the RLHF objective from Equation (4) can be written in closed-form $p_\theta^*(\mathbf{m}|\mathbf{p}) \propto p_{\text{ref}}(\mathbf{m}|\mathbf{p})\exp(\frac{1}{\beta}r(\mathbf{m},\mathbf{p}))$ [Peters and Schaal, 2007]. Following the preference optimization algorithm [Rafailov et al., 2023], we use the Bradley Terry (BT, [Bradley and Terry, 1952]) model $p(\mathbf{m}_1^0 \succ \mathbf{m}_2^0|\mathbf{p}) = \sigma(r(\mathbf{m}_1^0,\mathbf{p}) - r(\mathbf{m}_2^0,\mathbf{p}))$ to reformulate the RLHF objective as:

$$\mathcal{L}_{\text{DPO}}(\theta) = -\mathbb{E}_{(\mathbf{p},\mathbf{m}^w,\mathbf{m}^l)\sim\mathcal{D}}\left[\log\sigma\left(\beta\log\frac{p_\theta(\mathbf{m}_0^w|\mathbf{p})}{p_{\text{ref}}(\mathbf{m}_0^w|\mathbf{p})} - \beta\log\frac{p_\theta(\mathbf{m}_0^l|\mathbf{p})}{p_{\text{ref}}(\mathbf{m}_0^l|\mathbf{p})}\right)\right]. \tag{5}$$

Due to the intractability of $p_\theta(\mathbf{m}|\mathbf{p})$ for diffusion models, we instead follow recent work on diffusion-based preference optimization [Wallace et al., 2023] to align the whole reverse process and utilize Jensen's inequality to optimize its negative evidence lower bound optimization (ELBO):

$$\mathcal{L}_{\text{DPO-Diffusion}}(\theta) = -\mathbb{E}_{(\mathbf{p},\mathbf{m}_0^w,\mathbf{m}_0^l)\sim\mathcal{D},(\mathbf{m}_{1:T}^w,\mathbf{m}_{1:T}^l)\sim p_\theta}\left[\log\sigma\left(\beta\log\frac{p_\theta(\mathbf{m}_{0:T}^w)}{p_{\text{ref}}(\mathbf{m}_{0:T}^w)} - \beta\log\frac{p_\theta(\mathbf{m}_{0:T}^l)}{p_{\text{ref}}(\mathbf{m}_{0:T}^l)}\right)\right], \tag{6}$$

where we omit the conditioning on the protein target $\mathbf{p}$ for compactness. We further approximate the reverse process $p_\theta(\mathbf{m}_{1:T}|\mathbf{m}_0)$ with the forward process $q(\mathbf{m}_{1:T}|\mathbf{m}_0)$ for efficient sampling of $\mathbf{m}_{1:T}$, and obtain the following expression after some derivations [Wallace et al., 2023]:

$$\tilde{\mathcal{L}}_{\text{DPO-Diffusion}}(\theta) = -\mathbb{E}_{(\mathbf{p},\mathbf{m}_0^w,\mathbf{m}_0^l)\sim\mathcal{D},t\sim[0,T],\mathbf{m}_t^w\sim q,\mathbf{m}_t^l\sim q}\big[$$
$$\log\sigma\big(-\beta T\big(\mathbb{D}_{\text{KL}}(q(\mathbf{m}_{t-1}^w|\mathbf{m}_{0,t}^w)\|p_\theta(\mathbf{m}_{t-1}^w|\mathbf{m}_t^w)) - \mathbb{D}_{\text{KL}}(q(\mathbf{m}_{t-1}^w|\mathbf{m}_{0,t}^w)\|p_{\text{ref}}(\mathbf{m}_{t-1}^w|\mathbf{m}_t^w))$$
$$- \mathbb{D}_{\text{KL}}(q(\mathbf{m}_{t-1}^l|\mathbf{m}_{0,t}^l)\|p_\theta(\mathbf{m}_{t-1}^l|\mathbf{m}_t^l)) + \mathbb{D}_{\text{KL}}(q(\mathbf{m}_{t-1}^l|\mathbf{m}_{0,t}^l)\|p_{\text{ref}}(\mathbf{m}_{t-1}^l|\mathbf{m}_t^l))\big)\big)\big] \tag{7}$$

Let $[\hat{\mathbf{x}}_0,\hat{\mathbf{v}}_0]$ be the predicted atom position and type, which are fed into Equation (3) to obtain the posterior distributions. With the joint diffusion processes Equations (1) to (3) on both continuous $\mathbf{x}$ and discrete $\mathbf{v}$ features, the above KL divergences can be decomposed and calculated as:

$$\mathbb{D}_{\text{KL}}(q(\mathbf{m}_{t-1}|\mathbf{m}_{0,t})\|p(\mathbf{m}_{t-1}|\mathbf{m}_t)) = \mathbb{D}_{\text{KL}}^{\mathbf{x},t-1}(q(\mathbf{x}_{t-1}|\mathbf{x}_{0,t})\|p(\mathbf{x}_{t-1}|\mathbf{x}_t)) + \mathbb{D}_{\text{KL}}^{\mathbf{v},t-1}(q(\boldsymbol{c}_{t-1}|\boldsymbol{c}_{0,t})\|p(\boldsymbol{c}_{t-1}|\boldsymbol{c}_t)),$$

$$\mathbb{D}_{\text{KL}}^{\mathbf{x},t-1}(q(\mathbf{x}_{t-1}|\mathbf{x}_{0,t})\|p(\mathbf{x}_{t-1}|\mathbf{x}_t)) = \frac{1}{\tilde{\beta}_t}\|\tilde{\boldsymbol{\mu}}(\mathbf{x}_t,\mathbf{x}_0) - \tilde{\boldsymbol{\mu}}(\mathbf{x}_t,\hat{\mathbf{x}}_0)\|^2 + C = \gamma_t\|\mathbf{x}_0 - \hat{\mathbf{x}}_0\|^2 + C,$$

$$\mathbb{D}_{\text{KL}}^{\mathbf{v},t-1}(q(\boldsymbol{c}_{t-1}|\boldsymbol{c}_{0,t})\|p(\boldsymbol{c}_{t-1}|\boldsymbol{c}_t)) = \sum_k \tilde{\boldsymbol{c}}(\mathbf{v}_t,\mathbf{v}_0)_k\log\frac{\tilde{\boldsymbol{c}}(\mathbf{v}_t,\mathbf{v}_0)_k}{\tilde{\boldsymbol{c}}(\mathbf{v}_t,\hat{\mathbf{v}}_0)_k}, \tag{8}$$

where $\gamma_t = \frac{\bar{\alpha}_{t-1}\beta_t^2}{2\sigma_t^2(1-\bar{\alpha}_t)^2}$ and $C$ is a constant. Let $\hat{\mathbf{x}}_{0,\theta},\hat{\mathbf{v}}_{0,\theta}$ and $\hat{\mathbf{x}}_{0,\text{ref}},\hat{\mathbf{v}}_{0,\text{ref}}$ be the predictions from the fine-tuned and from the original pretrained model, respectively. Then, we can further obtain the preference optimization loss on $\mathbf{x}$ and $\mathbf{v}$, respectively, as follows:

$$\mathcal{L}_{t-1}^{\mathbf{x}}(\theta) = -\mathbb{E}\big[\log\sigma\big(-\beta T\gamma_t(\|\mathbf{x}_0^w - \hat{\mathbf{x}}_{0,\theta}^w\|^2 - \|\mathbf{x}_0^w - \hat{\mathbf{x}}_{0,\text{ref}}^w\|^2 - \|\mathbf{x}_0^l - \hat{\mathbf{x}}_{0,\theta}^l\|^2 + \|\mathbf{x}_0^l - \hat{\mathbf{x}}_{0,\text{ref}}^l\|^2)\big)\big]$$
$$\mathcal{L}_{t-1}^{\mathbf{v}}(\theta) = -\mathbb{E}\big[\log\sigma\big(-\beta T\big(\mathbb{D}_{\text{KL}}(\tilde{\boldsymbol{c}}(\mathbf{v}_t^w,\mathbf{v}_0^w)\|\tilde{\boldsymbol{c}}(\mathbf{v}_t^w,\hat{\mathbf{v}}_{0,\theta}^w)) - \mathbb{D}_{\text{KL}}(\tilde{\boldsymbol{c}}(\mathbf{v}_t^w,\mathbf{v}_0^w)\|\tilde{\boldsymbol{c}}(\mathbf{v}_t^w,\hat{\mathbf{v}}_{0,\text{ref}}^w))$$
$$- \mathbb{D}_{\text{KL}}(\tilde{\boldsymbol{c}}(\mathbf{v}_t^l,\mathbf{v}_0^l)\|\tilde{\boldsymbol{c}}(\mathbf{v}_t^l,\hat{\mathbf{v}}_{0,\theta}^l)) + \mathbb{D}_{\text{KL}}(\tilde{\boldsymbol{c}}(\mathbf{v}_t^l,\mathbf{v}_0^l)\|\tilde{\boldsymbol{c}}(\mathbf{v}_t^l,\hat{\mathbf{v}}_{0,\text{ref}}^l))\big)\big)\big] \tag{9}$$

With Jensen's inequality and the convexity of $-\log\sigma$, we can derive the final objective as a (weighted) sum of atom coordinate and type preference losses $\mathcal{L}_{t-1}^{\mathbf{x}} + \mathcal{L}_{t-1}^{\mathbf{v}}$, which turns the sum of the KL terms outside $-\log\sigma$ and serves as an upper bound of Equation (7):

$$\mathcal{L}_{\text{ALIDIFF}}(\theta) = -\mathbb{E}_{(\mathbf{p},\mathbf{m}_0^w,\mathbf{m}_0^l)\sim\mathcal{D},t\sim[0,T],\mathbf{m}_t^w\sim q,\mathbf{m}_t^l\sim q}\big[\mathcal{L}_{t-1}^{\mathbf{x}} + \mathcal{L}_{t-1}^{\mathbf{v}}\big] \geq \tilde{\mathcal{L}}_{\text{DPO-Diffusion}}(\theta), \tag{10}$$

where the preference is assigned separately to atom types $\mathbf{v}$ and coordinates $\mathbf{x}$. The loss decomposition imposes a fine-grained preference assignment on chemical elements and geometric strcutures and enables us to choose weights to balance the training of the two variables [Guan et al., 2023, 2024]. The overall training and sampling algorithms of ALIDIFF are summarized in Appendix B.

## 3.3 Exact Energy Optimization

Although DPO enjoys the advantage of efficient fine-tuning without fitting a reward function, recent theoretical investigations reveal that it is highly vulnerable to overfitting by pushing all the probability mass on the winning sample [Azar et al., 2024]. Specifically, the non-linear transformation $\log\sigma$

of Equation ([5]) pushes the $\log p_\theta(\mathbf{m}^w|\mathbf{p}) - \log p_\theta(\mathbf{m}^l|\mathbf{p})$ towards infinity, completely removing the likelihood for the losing sample regardless of any regularization in the original RLHF setup Equation ([4]) [Azar et al., 2024, Tang et al., 2024]. Let us analyze the problem with an example consisting of two ligand molecules $\mathbf{m}^w$ and $\mathbf{m}^l$ with their rewards measured as $\mathbf{r}^w$ and $\mathbf{r}^l$ (*e.g.*, calculated from binding energy). The DPO objective in Equation ([10]) tends to just greedily maximize towards $p(\mathbf{m}^w \succ \mathbf{m}^l|\mathbf{p}) \to 1$. However, the optimal preference probability can be calculated by the BT model [Bradley and Terry, 1952] as $\hat{p}(\mathbf{m}^w \succ \mathbf{m}^l|\mathbf{p}) = \sigma(\mathbf{r}^w - \mathbf{r}^l)$, and our alignment goal is to shift the distribution to align with this $\hat{p}$ instead of greedy maximization. To address the over-optimization issue, we introduce an improved objective with regularization on preference maximization, named Exact Energy Preference Optimization (E²PO). Let $\bar{\mathcal{L}}_t^{\mathbf{x}}(\theta)$ and $\bar{\mathcal{L}}_t^{\mathbf{v}}(\theta)$ denote terms for reverse preference optimization:

$$\bar{\mathcal{L}}_{t-1}^{\mathbf{x}}(\theta) = 1 - \mathcal{L}_{t-1}^{\mathbf{x}}(\theta), \quad \bar{\mathcal{L}}_{t-1}^{\mathbf{v}}(\theta) = 1 - \mathcal{L}_{t-1}^{\mathbf{v}}(\theta). \tag{11}$$

Our E²PO objective function takes a cross-entropy form to align the distributions $p_\theta(\mathbf{m}^w \succ \mathbf{m}^l|\mathbf{p})$ towards $\hat{p}(\mathbf{m}^w \succ \mathbf{m}^l|\mathbf{p})$. Formally, it is given by:

$$\mathcal{L}_{\text{ALIDIFF-E}^2\text{PO}}(\theta) = -\mathbb{E}_{(\mathbf{p},\mathbf{m}_0^w,\mathbf{m}_0^l)\sim\mathcal{D},t\sim[0,T],\mathbf{m}_t^w\sim q,\mathbf{m}_t^l\sim q}\Big[$$
$$(\sigma(\mathbf{r}^w - \mathbf{r}^l))(\mathcal{L}_{t-1}^{\mathbf{x}} + \mathcal{L}_{t-1}^{\mathbf{v}}) + (1 - \sigma(\mathbf{r}^w - \mathbf{r}^l))(\bar{\mathcal{L}}_{t-1}^{\mathbf{x}} + \bar{\mathcal{L}}_{t-1}^{\mathbf{v}})\Big], \tag{12}$$

where the second term $\bar{\mathcal{L}}_{t-1}^{\mathbf{x}} + \bar{\mathcal{L}}_{t-1}^{\mathbf{v}}$ weighted by $1 - \sigma(\mathbf{r}^w - \mathbf{r}^l)$ helps to alleviate the overfitting on the winning data sample. Notably, for $\mathbf{r}^w >> \mathbf{r}^l$, we have $\sigma(\mathbf{r}^w - \mathbf{r}^l) \approx 1$, indicating that the regularized objective in Equation ([12]) will still change back to the original objective in Equation ([10]), where overfitting on the extremely better data is expected. In principle, with the regularization objective, we have:

**Theorem 3.1.** *The objective function in Equation* ([12]) *optimizes a variational upper bound of the KL-divergence* $\mathbb{D}_{\text{KL}}(\hat{p}^*(\mathbf{m}|\mathbf{p})||\hat{p}_\theta(\mathbf{m}|\mathbf{p}))$, *where* $\hat{p}^*(\mathbf{m}|\mathbf{p}) \propto p_{\text{ref}}(\mathbf{m}|\mathbf{p})\exp(r(\mathbf{m},\mathbf{p}))$ *and* $\hat{p}_\theta(\mathbf{m}|\mathbf{p}) \propto p_{\text{ref}}(\mathbf{m}|\mathbf{p})\left(\frac{p_\theta(\mathbf{m}|\mathbf{p})}{p_{\text{ref}}(\mathbf{m}|\mathbf{p})}\right)^\beta$.

The theorem provides an analytical guarantee for the optimal shifted distribution after alignment that avoids over-optimization. Assuming we achieve convergence on the KL divergence, we have that $p_{\text{ref}}(\mathbf{m}|\mathbf{p})\exp(r(\mathbf{m},\mathbf{p})) \propto p_{\text{ref}}^{1-\beta}(\mathbf{m}|\mathbf{p})p_\theta^\beta(\mathbf{m}|\mathbf{p})$ which further gives us $p_\theta(\mathbf{m}|\mathbf{p}) \propto p_{\text{ref}}(\mathbf{m}|\mathbf{p})\exp(\frac{1}{\beta}r(\mathbf{m},\mathbf{p}))$, where a smaller $\beta$ encourages a sharper shift towards the user-defined reward function. We give the full derivations in Appendix C, and analyze the empirical effect on generation quality in Section [4].

## 4 Experiment

### 4.1 Experiment Setup

**Dataset**. We train and evaluate ALIDIFF using the CrossDocked2020 dataset [Francoeur et al., 2020]. Following the common setup in this field [Luo et al., 2021, Guan et al., 2023], we refined the initial 22.5 million docked protein binding complexes by selecting docking poses with RMSD lower than 1Å with the ground truth and diversifying proteins with a sequence identity below 30%. To apply ALIDIFF, we further preprocess our data and construct a dataset of the form $\mathcal{D} = \{(\mathbf{p},\mathbf{m}^w,\mathbf{m}^l)\}$, where $\mathbf{p}$ denotes the protein, $\mathbf{m}^w$ denotes the preferred molecules, and $\mathbf{m}^l$ denotes rejected molecules based on the user-defined reward. In our setting, we choose two ligand molecules per pocket site and label the preference by a certain reward, *e.g.* binding energy for our main benchmark. We provide ablations with more reward functions in Section [4.3]. Details of preference pair selection are presented in Appendix E. The final dataset uses a train and test split of 65K and 100.

**Baselines**. We compare our model with the following baselines: liGAN [Ragoza et al., 2022] is a conditional VAE model that utilizes a 3D CNN architecture to both encode and generate voxelized representations of atomic densities; AR [Luo et al., 2021], Pocket2Mol [Peng et al., 2022] and GraphBP [Liu et al., 2022] are autoregressive models that learn graph neural networks to generate 3D molecules atom by atom sequentially; TargetDiff [Guan et al., 2023] and DecompDiff [Guan et al., 2024] are diffusion-based approaches for generating atomic coordinates and types via a joint denoising process; IPDiff [Huang et al., 2023] is the most recent state-of-the-art diffusion-based

Table 1: Summary of binding affinity and molecular properties of reference molecules and molecules generated by ALIDIFF and baselines. (↑) / (↓) denotes whether a larger / smaller number is preferred. Top 2 results are bolded and underlined, respectively.

| Methods | Vina Score (↓) | | Vina Min (↓) | | Vina Dock (↓) | | High Affinity(↑) | | QED(↑) | | SA(↑) | | Diversity(↑) | |
| --- | --- | --- | --- | --- | --- | --- | --- | --- | --- | --- | --- | --- | --- | --- |
| | Avg. | Med. | Avg. | Med. | Avg. | Med. | Avg. | Med. | Avg. | Med. | Avg. | Med. | Avg. | Med. |
| liGAN* | - | - | - | - | -6.33 | -6.20 | 21.1% | 11.1% | 0.39 | 0.39 | 0.59 | 0.57 | 0.66 | 0.67 |
| GraphBP* | - | - | - | - | -4.80 | -4.70 | 14.2% | 6.7% | 0.43 | 0.45 | 0.49 | 0.48 | **0.79** | **0.78** |
| AR | -5.75 | -5.64 | -6.18 | -5.88 | -6.75 | -6.62 | 37.9% | 31.0% | 0.51 | 0.50 | 0.63 | 0.63 | 0.70 | 0.70 |
| Pocket2Mol | -5.14 | -4.70 | -6.42 | -5.82 | -7.15 | -6.79 | 48.4% | 51.0% | **0.56** | **0.57** | **0.74** | **0.75** | 0.69 | 0.71 |
| TargetDiff | -5.47 | -6.30 | -6.64 | -6.83 | -7.80 | -7.91 | 58.1% | 59.1% | 0.48 | 0.48 | 0.58 | 0.58 | 0.72 | 0.71 |
| DecompDiff | -5.67 | -6.04 | -7.04 | -7.09 | -8.39 | -8.43 | 64.4% | 71.0% | 0.45 | 0.43 | 0.61 | 0.60 | 0.68 | 0.68 |
| IPDiff | -6.42 | -7.01 | -7.45 | -7.48 | -8.57 | -8.51 | 69.5% | 75.5% | 0.52 | 0.53 | 0.61 | 0.59 | 0.74 | 0.73 |
| **ALIDIFF** | **-7.07** | **-7.95** | **-8.09** | **-8.17** | **-8.90** | **-8.81** | **73.4%** | **81.4%** | 0.50 | 0.50 | 0.57 | 0.56 | 0.73 | 0.71 |
| Reference | -6.36 | -6.41 | -6.71 | -6.49 | -7.45 | -7.26 | - | - | 0.48 | 0.47 | 0.73 | 0.74 | - | - |

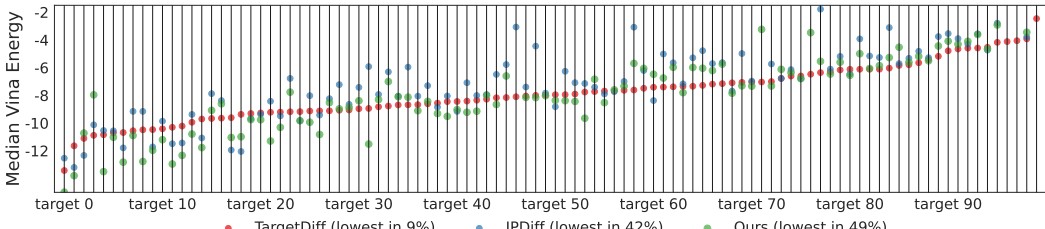

Figure 3: Median Vina energy for different generated molecules (TargetDiff, IPDiff, ALIDIFF) across 100 testing samples, sorted by the median Vina energy of molecules generated from ALIDIFF.

approach that further integrates the interactions between the target protein and the molecular ligand into the generation process.

**Evaluation metrics.** We evaluate the generated molecules by comparing *binding affinity* with the target and critical *molecular properties*. We analyze the generated molecules across 100 test proteins, reporting the mean and median for affinity-based metrics (Vina Score, Vina Min, Vina Dock, and High Affinity) and molecular property metrics (drug-likeness QED [Bickerton et al., 2012], synthesizability SA [Ertl and Schuffenhauer, 2009], and diversity). We use AutoDock Vina [Eberhardt et al., 2021] to estimate binding affinity scores, using the common setup described by Luo et al. [2021], Ragoza et al. [2022]. Specifically, Vina Score estimates binding affinity from the generated 3D structures, Vina Min refines the structure through local minimization before estimation, Vina Dock uses a re-docking procedure to reflect the optimal binding affinity, and High Affinity gauges the percentage of generated molecules that bind better than reference molecules per protein.

## 4.2 Results

**Binding Affinity and Molecular Properties.** We compare the performance of our proposed method ALIDIFF against the above baseline methods. Our model is fine-tuned from IPDiff, the ligand generative model. We report the results in Table 1, and leave more implementation details in Appendix D. As shown in the results, ALIDIFF significantly outperforms all non-diffusion-based models in binding-related metrics, and also surpasses our base model IPDiff in all binding affinity related metrics by a notable margin. In particular, ALIDIFF increases the binding-related metrics Avg. Vina Score, Vina Min, and Vina Dock by 10.1%, 8.56%, and 3.9% compared with IPDiff. Our superior performance in binding-related metrics demonstrates the effectiveness of energy preference

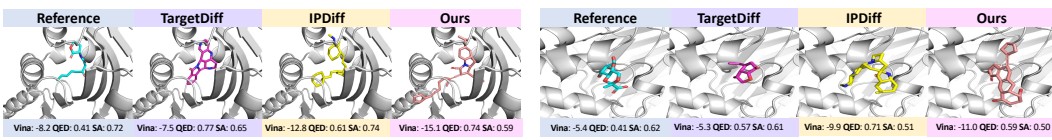

Figure 4: Visualizations of reference molecules and generated ligands for protein pockets (1l31, 2e24) generated by TargetDiff, IPDiff, and ALIDIFF. Vina score, QED, and SA are reported below.

Table 2: Effect of combining multiple reward objectives. Affinity denotes ALIDIFF, whereas Affinity+SA denotes combining both synthetic accessibility and affinity as reward function.

| Choice of reward | Vina Score ($\downarrow$) | | Vina Min ($\downarrow$) | | Vina Dock ($\downarrow$) | | High Affinity($\uparrow$) | | QED($\uparrow$) | | SA($\uparrow$) | | Diversity($\uparrow$) | |
|---|---|---|---|---|---|---|---|---|---|---|---|---|---|---|
| | Avg. | Med. | Avg. | Med. | Avg. | Med. | Avg. | Med. | Avg. | Med. | Avg. | Med. | Avg. | Med. |
| Affinity | -7.07 | -7.95 | **-8.09** | **-8.17** | **-8.90** | **-8.81** | **73.4%** | **81.4%** | 0.50 | 0.50 | 0.57 | 0.56 | 0.73 | 0.71 |
| Affinity+SA | -6.87 | -7.76 | -8.00 | -8.08 | -8.81 | -8.72 | 72.7% | 80.8% | **0.52** | **0.55** | **0.60** | **0.59** | **0.74** | **0.73** |
| Affinity+QED | **-7.11** | **-8.02** | -8.01 | -7.99 | -8.17 | -8.72 | 73.7% | 82.0% | 0.51 | 0.52 | 0.57 | 0.57 | 0.73 | **0.73** |

Table 3: Comparison of DPO and E$^2$PO with pretrained and supervised fine-tuned models. ALIDIFF with DPO takes energy ranking, and with E$^2$PO uses exact energy for preference optimization.

| Methods | Vina Score ($\downarrow$) | | Vina Min ($\downarrow$) | | Vina Dock ($\downarrow$) | | High Affinity($\uparrow$) | | QED($\uparrow$) | | SA($\uparrow$) | | Diversity($\uparrow$) | |
|---|---|---|---|---|---|---|---|---|---|---|---|---|---|---|
| | Avg. | Med. | Avg. | Med. | Avg. | Med. | Avg. | Med. | Avg. | Med. | Avg. | Med. | Avg. | Med. |
| IPDiff | -6.42 | -7.01 | -7.45 | -7.48 | -8.57 | -8.51 | 69.5% | 75.5% | **0.52** | **0.53** | **0.61** | **0.59** | **0.74** | **0.73** |
| IPDiff$_{SFT}$ | -6.53 | -6.62 | -7.27 | -7.09 | -8.14 | -8.09 | 67.5% | 72.5% | 0.48 | 0.48 | **0.61** | **0.59** | 0.72 | 0.69 |
| **ALIDIFF-DPO** | -6.81 | -7.62 | -7.75 | -7.79 | -8.58 | -8.55 | 69.7% | 71.1% | 0.50 | 0.51 | 0.56 | 0.56 | **0.74** | 0.72 |
| **ALIDIFF-E$^2$PO** | **-7.07** | **-7.95** | **-8.09** | **-8.17** | **-8.90** | **-8.81** | **73.4%** | **81.4%** | 0.50 | 0.50 | 0.57 | 0.56 | 0.73 | 0.71 |

optimization. Figure 3 shows the median Vina energy of the proposed model, compared with TargetDiff and IPDiff, two diffusion-based state-of-the-art models in target-aware molecule generation. We observe that ALIDIFF surpasses these baseline models and generates molecules with the highest binding affinity for 49% of the protein targets in the test set. In property-related metrics, we observe only a slight decrease in QED, SA, and diversity, compared with IPDiff. Specifically, with approximately 10.1% improvement on Avg. Vina Score, we observe a minor decrease in Avg. SA (-6.5%), Avg. QED (-3.8%), and diversity(-1.4%). Figure 4 presents examples of ligand molecules generated by ALIDIFF, TargetDiff, and IPDiff. The figure shows that our generated molecules maintain reasonable structures and high binding affinity compared with all baselines, indicating their potential as promising candidate ligands. Additional experimental results and visualized examples of these molecules are in Appendices E and F.

We also notice a trade-off between binding affinity and property-related metrics. While we achieve state-of-the-art performance on all binding affinity metrics, the performance on QED and SA metrics slightly decreases. This phenomenon has been commonly observed in previous studies where achieving high binding affinity can often sacrifice other molecular metrics [Guan et al., 2023, Huang et al., 2023]. This is because the highest affinity can potentially only be achieved by rather specific and unique molecules, which are harder to synthesize than simple molecules, and hence these trade-offs are expected. Besides, in real-world drug discovery, binding affinity is typically a more critical metric as molecules with more stable interaction with the pocket site are important, whereas QED and SA work mainly as rough filters [Guan et al., 2023]. For these reasons, we believe the deterioration in molecular properties is well compensated by the improvement in binding affinity, especially with such little deterioration in property metrics. In addition, in the following section (Table 2), we further discuss incorporating molecular properties into the reward, which shows slightly lower performance gain on affinity but archives improvements also on molecular properties.

## 4.3 Ablation Studies

**Effect of reward objectives.** To further explore the potential of ALIDIFF, we evaluate the effect of combining optimization objectives ($\mathbf{r} = \mathbf{r}_{affinity} + \mathbf{r}_{SA}$; $\mathbf{r} = \mathbf{r}_{affinity} + \mathbf{r}_{QED}$) and investigate whether such a combined reward function can lead to better molecular properties to counter the trade-off we discussed before. As shown in Table 2, the results indicate that finetuning solely with binding affinity apparently achieves better performance in terms of binding affinity metrics. However, ALIDIFF-Affinity+SA generates compounds with better drug-likeness (QED) and synthetic accessibility (SA). Both models exhibit similar performance in terms of structural diversity. This suggests that while ALIDIFF-Affinity is superior for binding affinity, incorporating synthetic accessibility considerations (Affinity + SA) results in compounds that are more drug-like and easier to synthesize, enabling more efficient multi-objective drug development. Moreover, ALIDIFF-Affinity+QED achieves better binding affinity compared with ALIDIFF-Affinity, while the improvement in QED is relatively minimal. Thus, balancing these objectives highlights the potential for overcoming trade-offs in molecular optimization.

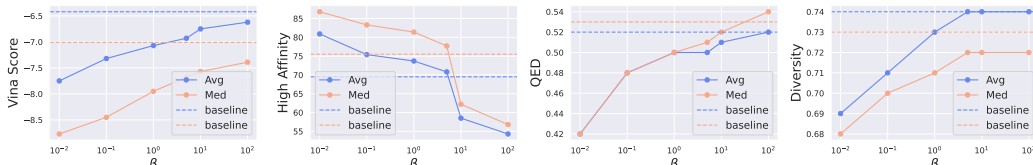

Figure 5: Ablation analysis of ALIDIFF under different $\beta$. Vina Score, High Affinity, QED, and diversity are reported, where blue lines represent ALIDIFF-DPO, and orange lines represent ALIDIFF. The dotted lines represent the baseline IPDiff.

**Comparison with Supervised Fine-Tuning.** Supervised Fine-Tuning (SFT) serves as an alternative method for generating molecules with user-defined optimization objectives. We select the top 50% protein-ligand samples with higher quality in user-defined reward from the training dataset and fine-tune the baseline model with the same training and sampling setting. The results in Table 3 show that SFT did not show improvement over the baseline, and ALIDIFF demonstrates significantly superior results compared to SFT.

**Effect of preference optimization methods.** As discussed in Section 3.3, the original DPO objective is vulnerable to overfitting and we propose to avoid it with regularization by weighting preference losses with the user-defined rewards. We compare the direct use of energy preference optimization by ranking molecule pairs (ALIDIFF-DPO) and exact energy optimization with user-defined reward function (ALIDIFF-E$^2$PO) in Table 3. The results show that ALIDIFF-E$^2$PO achieves superior performance over ALIDIFF-DPO in all binding affinity metrics (Vina Score, Vina Min, and Vina Dock) while maintaining competitive scores in QED, SA and diversity. In terms of drug-likeness and structural diversity, ALIDIFF-E$^2$PO performs competitively, indicating that while it prioritizes binding affinity, it still maintains favorable drug-like properties and diversity. This further supports our previous hypothesis regarding the trade-off between binding affinity and molecular properties. An additional ablation study on the effect of exact energy optimization is presented in Appendix E.

**General applicability to ligand diffusion models.** We further justify the general applicability of the proposed approach by finetuning another diffusion-based SBDD model, TargetDiff [Guan et al., 2023], with exact energy optimization (ALIDIFF-T), As shown in Table 4, ALIDIFF-T surpasses TargetDiff on all binding affinity and molecular properties, with a 6.2%, 16.6%, 6.9%, 2.8% increase in Avg. Vina Score, QED, SA, and diversity, respectively. The results further justify that our approach is generally applicable to diffusion-based SBDD models. Notably, ALIDIFF-T archives even better QED and SA compared with ALIDIFF, which allows users to choose the model based on the specific purpose for molecular properties. Also, we notice the percentage of improvement of binding affinity from Target-

Table 4: Finetuning TargetDiff with ALIDIFF. ALIDIFF-T denotes our fine-tuned model with the same reward objective on TargetDiff.

| Metric | TargetDiff | | ALIDIFF-T | |
|---|---|---|---|---|
| | Avg. | Med. | Avg. | Med. |
| Vina Score | -5.47 | -6.30 | **-5.81** | **-6.51** |
| Vina Min | -6.64 | -6.83 | **-6.94** | **-7.01** |
| Vina Dock | -7.80 | -7.91 | **-7.92** | **-7.97** |
| QED | 0.48 | 0.48 | **0.56** | **0.56** |
| SA | 0.58 | 0.58 | **0.62** | **0.60** |
| Diversity | 0.72 | 0.71 | **0.74** | **0.75** |

Diff to ALIDIFF-T is slightly lower than that from IPDiff to ALIDIFF. This can be explained as preference optimization is more effective when the model distribution is more similar to the preference data distribution, and IPDiff is shown to fit CrossDocked data better than TargetDiff [Huang et al., 2023].

**Strength of $\beta$.** We further evaluate ligand molecules generated by ALIDIFF trained with varying $\beta$ values in Figure 5. Recall that $\beta$ influences the scale of energy preference optimization and regularization with respect to the reference model. The results indicate a clear trade-off between binding affinity and molecular properties with varying $\beta$. Lower $\beta$ values (e.g., 0.01) significantly enhance binding affinity metrics (Vina Score, Vina Min, Vina Dock), but at the cost of lower drug-likeness (QED) and diversity. Conversely, higher $\beta$ values improve QED, suggesting that these configurations generate more drug-like compounds while maintaining consistent synthetic accessibility and diversity. We believe $\beta$ reaches an equilibrium around $\beta = 1$, where binding affinity is maximized without sacrificing too much loss in molecular properties. This ablation study demonstrates that the parameter $\beta$ can offer a useful tool to train ALIDIFF models with different desired trade-offs between binding affinity and useful molecular properties, which can vary for different drug development use cases.

# 5 Conclusion

In this paper, we present ALIDIFF, a novel framework to align pretrained target-aware molecule diffusion models with desired functional properties via preference optimization. Our key innovation is the Exact Energy Preference Optimization method, which enables efficient and exact alignment of the diffusion model towards regions of lower binding energy and structural rationality specified by user-defined reward functions. Extensive experiments on the CrossDocked2020 benchmark demonstrate the strong performance of ALIDIFF. By incorporating user-defined reward functions and an improved Exact Energy Preference Optimization method, ALIDIFF successfully achieves state-of-the-art performance in binding affinity while maintaining competitive molecular properties. In the future, we plan to explore more expressive molecular reward function classes within our framework and extend ALIDIFF to real-world prospective drug design settings by integrating it into online drug discovery pipelines.

## Acknowledgement

We thank Jiaqi Han for the discussions on this project. We gratefully acknowledge the support of ARO (W911NF-21-1-0125), ONR (N00014-23-1-2159), NVIDIA, and Chan Zuckerberg Biohub. We also gratefully acknowledge the support of NSF under Nos. OAC-1835598 (CINES), CCF-1918940 (Expeditions), DMS-2327709 (IHBEM), IIS-2403318 (III); Stanford Data Applications Initiative, Wu Tsai Neurosciences Institute, Stanford Institute for Human-Centered AI, Chan Zuckerberg Initiative, Amazon, Genentech, GSK, Hitachi, SAP, and UCB. Minkai Xu thanks the generous support of Sequoia Capital Stanford Graduate Fellowship.

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

## A Limitations and Future Work

While ALIDIFF exhibits promising performance, there are still potential limitations to our current approach. For example, ALIDIFF takes binding affinity as our reward function which is computed by AutoDock Vina [Eberhardt et al., 2021] in this work. However, computing binding energy via software is an approximation and sometimes can be very inaccurate. In the future, we plan to explore experiment-measured energy or ensemble different binding affinity calculation software, *e.g.*, GlideScore [Friesner et al., 2004] In addition, in this work, we focus on an offline learning setting where the preference pairs are off-the-shelf. This is because computing binding affinity is computationally expensive. An important future direction to extend the work toward real-world drug discovery scenarios could be incorporating the online setting but with a limited number of query.

## B Algorithm

The pseudo-code for ALIDIFF and ALIDIFF-T are provided below. Sampling procedures are the same as Guan et al. [2023] and Huang et al. [2023].

---

**Algorithm 1** Training Procedure ALIDIFF

---

1: **Input:** Protein-ligand binding dataset $\{\mathcal{P}, \mathcal{M}^w, \mathcal{M}^l\}_1^N$, pre-trained neural network $\phi_\theta$, reference network $\phi_{\text{ref}}$, learnable neural network $\psi_{\theta 2}$ and pretrained interaction prior network $\psi_{\text{IP}}$.
2: **while** $\phi_\theta$ and $\psi_{\theta 2}$ not converge **do**
3:      $[\![\mathbf{p}_0, \mathbf{m}_0^w, \mathbf{m}_0^l]\!] \sim \{\mathcal{P}, \mathcal{M}^w, \mathcal{M}^l\}_{i=1}^N$ where $\mathbf{m}_0^w = \{\mathbf{x}_0^w, \mathbf{v}_0^w\}, \mathbf{m}_0^l = \{\mathbf{x}_0^l, \mathbf{v}_0^l\}$
4:      Obtain $\mathbf{r}^w, \mathbf{r}^l$ for $\mathbf{m}^w, \mathbf{m}^l$, respectively.
5:      $t \sim U(0, \ldots, T)$
6:      Move the complex to make CoM of protein atoms zero
7:      Obtain shifts $[\![\mathbf{s}_0^{\mathcal{M}_w}, \mathbf{s}_0^{\mathcal{M}_l}]\!]$ and interactions $[\![\mathbf{f}_0^{\mathcal{M}_w}, \mathbf{f}_0^{\mathcal{M}_l}, \mathbf{f}_0^{\mathcal{P}}]\!]$ from $\psi_{\text{IP}}$ and $\psi_{\theta 2}$ according to [Huang et al., 2023].
8:      Perturb $\mathbf{x}_0^w, \mathbf{x}_0^l$ to obtain $\mathbf{x}_t^w, \mathbf{x}_t^l$ with shifts $\mathbf{s}_0^{\mathcal{M}_w}, \mathbf{s}_0^{\mathcal{M}_l}$
9:        $\epsilon \sim \mathcal{N}(0, \mathbf{I})$
10:        $\mathbf{x}_t^w = \sqrt{\bar{\alpha}_t}\mathbf{x}_0^w + \mathbf{s}_t^{\mathcal{M}_w} + \sqrt{1 - \bar{\alpha}_t}\epsilon, \mathbf{x}_t^l = \sqrt{\bar{\alpha}_t}\mathbf{x}_0^l + \mathbf{s}_t^{\mathcal{M}_l} + \sqrt{1 - \bar{\alpha}_t}\epsilon$
11:      Perturb $\mathbf{v}_0^w, \mathbf{v}_0^l$ to obtain $\mathbf{v}_t^w, \mathbf{v}_t^l$
12:        $g \sim \text{Gumbel}(0, 1)$
13:        $\log c^w = \log(\bar{\alpha}_t\mathbf{v}_0^w + (1 - \bar{\alpha}_t/K), \log c^l = \log(\bar{\alpha}_t\mathbf{v}_0^l + (1 - \bar{\alpha}_t/K)$
14:        $\mathbf{v}_t^w = \text{onehot}(\arg\max_i(g_i + \log c_i^w)), \mathbf{v}_t^l = \text{onehot}(\arg\max_i(g_i + \log c_i^l))$
15:      Embed $\mathbf{v}_t^w, \mathbf{v}_t^l$ into $\tilde{\mathbf{h}}_t^{w,0}, \tilde{\mathbf{h}}_t^{\mathcal{M}_l,0}$, and embed $\mathbf{v}_0^{\mathcal{P}}$ into $\tilde{\mathbf{h}}_t^{\mathcal{P},0}$
16:      Obtain features $[\![\mathbf{h}_t^{w,0}, \mathbf{h}_t^{\mathcal{M}_l,0}, \mathbf{h}_t^{\mathcal{P},0}]\!]$ through prior-conditioning
17:      Predict $(\hat{\mathbf{x}}_{0|t}^w, \hat{\mathbf{v}}_{0|t}^w)$ from $\phi_\theta([[\mathbf{h}_t^{\mathcal{M}_w,0}, \mathbf{h}_t^{\mathcal{P},0}]], [\![\mathbf{f}_0^{\mathcal{M}_w}, \mathbf{f}_0^{\mathcal{P}}]\!])$
18:      Predict $(\hat{\mathbf{x}}_{0|t}^l, \hat{\mathbf{v}}_{0|t}^l)$ from $\phi_\theta([\![\mathbf{h}_t^{\mathcal{M}_l,0}, \mathbf{h}_t^{\mathcal{P},0}]\!], [\![\mathbf{f}_0^{\mathcal{M}_l}, \mathbf{f}_0^{\mathcal{P}}]\!])$
19:      Predict $(\hat{\mathbf{x}}_{0|t,\text{ref}}^w, \hat{\mathbf{v}}_{0|t,\text{ref}}^w)$ from $\phi_{\text{ref}}([\![\mathbf{h}_t^{\mathcal{M}_w,0}, \mathbf{h}_t^{\mathcal{P},0}]\!], [\![\mathbf{f}_0^{\mathcal{M}_w}, \mathbf{f}_0^{\mathcal{P}}]\!])$
20:      Predict $(\hat{\mathbf{x}}_{0|t,\text{ref}}^l, \hat{\mathbf{v}}_{0|t,\text{ref}}^l$ from $\phi_{\text{ref}}([\![\mathbf{h}_t^{\mathcal{M}_l,0}, \mathbf{h}_t^{\mathcal{P},0}]\!], [\![\mathbf{f}_0^{\mathcal{M}_l}, \mathbf{f}_0^{\mathcal{P}}]\!])$
21:      Compute loss $L$ with $(\hat{\mathbf{x}}_{0|t}^w, \hat{\mathbf{v}}_{0|t}^w), (\mathbf{x}_0^l, \mathbf{v}_0^l), (\hat{\mathbf{x}}_{0|t,\text{ref}}^w, \hat{\mathbf{v}}_{0|t,\text{ref}}^w), (\hat{\mathbf{x}}_{0|t,\text{ref}}^l, \text{and } \hat{\mathbf{v}}_{0|t,\text{ref}}^l)$
       according to Equation (12)
22:      Update $\theta$ and $\theta 2$ by minimizing $L$
23: **end while**

---

**Algorithm 2** Training Procedure for ALIDIFF-T

---
1: **Input:** Protein-ligand binding dataset $\{\mathcal{P}, \mathcal{M}^w, \mathcal{M}^l\}_1^N$, pre-trained neural network $\phi_\theta$, reference network $\phi_{\text{ref}}$
2: **while** $\phi_\theta$ not converge **do**
3: $\quad [\![\mathbf{p}, \mathbf{m}_0^w, \mathbf{m}_0^l]\!] \sim \{\mathcal{P}, \mathcal{M}^w, \mathcal{M}^l\}_{i=1}^N$ where $\mathbf{m}_0^w = \{\mathbf{x}_0^w, \mathbf{v}_0^w\}, \mathbf{m}_0^l = \{\mathbf{x}_0^l, \mathbf{v}_0^l\}$
4: $\quad$ Sample diffusion time $t \sim U(0, \dots, T)$
5: $\quad$ Move the complex to make CoM of protein atoms zero
6: $\quad$ Perturb $\mathbf{x}_0^w, \mathbf{x}_0^l$ to obtain $\mathbf{x}_t^w, \mathbf{x}_t^l$: $\mathbf{x}_t^w = \sqrt{\alpha_t \mathbf{x}_0^w + (1-\alpha_t)\epsilon}, \mathbf{x}_t^l = \sqrt{\alpha_t \mathbf{x}_0^l + (1-\alpha_t)\epsilon}$, where $\epsilon \sim \mathcal{N}(0, I)$
7: $\quad$ Perturb $\mathbf{v}_0^w, \mathbf{v}_0^l$ to obtain $v_t^w, v_t^l$:
8: $\quad\quad logc^w = log(\alpha_t \mathbf{v}_0^w + (1-\alpha_t)/K)$
9: $\quad\quad logc^l = log(\alpha_t \mathbf{v}_0^l + (1-\alpha_t)/K)$
10: $\quad\quad \mathbf{v}_t^w = \text{one\_hot}(\arg\max[g_i + logc_i^w])$
11: $\quad\quad \mathbf{v}_t^l = \text{one\_hot}(\arg\max[g_i + logc_i^l])$, where $g \sim \text{Gumbel}(0, 1)$
12: $\quad$ Predict $[\hat{\mathbf{x}}_0^w, \hat{\mathbf{v}}_0^w]$ from $[\mathbf{x}_t^w, \mathbf{v}_t^w]$ with $\phi_\theta$: $[\hat{\mathbf{x}}_0^w, \hat{\mathbf{v}}_0^w] = \phi_\theta([\mathbf{x}_t^w, \mathbf{v}_t^w], t, \mathbf{p})$
13: $\quad$ Predict $[\hat{\mathbf{x}}_0^l, \hat{\mathbf{v}}_0^l]$ from $[\mathbf{x}_t^l, \mathbf{v}_t^l]$ with $\phi_\theta$: $[\hat{\mathbf{x}}_0^l, \hat{\mathbf{v}}_0^l] = \phi_\theta([\mathbf{x}_t^l, \mathbf{v}_t^l], t, \mathbf{p})$
14: $\quad$ Predict $[\hat{\mathbf{x}}_0^w, \bar{\mathbf{v}}_0^w]$ from $[\mathbf{x}_t^w, \mathbf{v}_t^w]$ with $\phi_{\text{ref}}$: $[\hat{\mathbf{x}}_0^w, \hat{\mathbf{v}}_0^w] = \phi_{\text{ref}}([\mathbf{x}_t^w, \mathbf{v}_t^w], t, \mathbf{p})$
15: $\quad$ Predict $[\bar{\mathbf{x}}_{0,\text{ref}}^l, \bar{\mathbf{v}}_{0,\text{ref}}^l]$ from $[\mathbf{x}_t^l, \mathbf{v}_t^l]$ with $\phi_{\text{ref}}$: $[\bar{\mathbf{x}}_{0,\text{ref}}^l, \bar{\mathbf{v}}_{0,\text{ref}}^l] = \phi_{\text{ref}}([\mathbf{x}_t^l, \mathbf{v}_t^l], t, \mathbf{p})$
16: $\quad$ Compute $\mathcal{L}(\theta) = \mathcal{L}_\mathbf{x}(\theta) + \alpha\mathcal{L}_\mathbf{v}(\theta)$ according to Equation (10)
17: $\quad$ Update $\theta$ by minimizing $L$
18: **end while**

---

## C  Proof

**Theorem 3.1.** *The objective function Equation* (12) *optimizes a variational upper bound of the KL-divergence* $\mathbb{D}_{\text{KL}}\big(\hat{p}^*(\mathbf{m}|\mathbf{p})||\hat{p}_\theta(\mathbf{m}|\mathbf{p})\big)$, *where* $\hat{p}^*(\mathbf{m}|\mathbf{p}) \propto p_{\text{ref}}(\mathbf{m}|\mathbf{p})\exp(r(\mathbf{m}, \mathbf{p}))$ *and* $\hat{p}_\theta(\mathbf{m}|\mathbf{p}) \propto p_{\text{ref}}(\mathbf{m}|\mathbf{p})\left(\frac{p_\theta(\mathbf{m}|\mathbf{p})}{p_{\text{ref}}(\mathbf{m}|\mathbf{p})}\right)^\beta$.

We prove the theorem with Lemmas C.1 and C.2. Lemma C.1 justifies the least square objective is the variational upper bound for preference optimization, and Lemma C.2 shows that regularized preference optimization corresponds to exact KL divergences between the optimal and parameterized distributions. A version of similar proof can be found in Wallace et al. [2023] and Ji et al. [2024], Chen et al. [2024] respectively, and to be self-contained we incorporate these proofs here. Compared with Wallace et al. [2023], we introduce an additional term into the diffusion optimization. And compared with Ji et al. [2024], Chen et al. [2024], we explicitly drop the assumption for drawing infinite samples $\mathbf{m}$ for each pocket $\mathbf{p}$.

**Lemma C.1.** *The objective function Equation* (12) $\mathcal{L}_{\text{ALIDIFF-}E^2PO}(\theta) = -\mathbb{E}_{(\mathbf{p}, \mathbf{m}_0^w, \mathbf{m}_0^l)\sim\mathcal{D}, t\sim[0,T], \mathbf{m}_t^w\sim q, \mathbf{m}_t^l\sim q}\big[(\sigma(\mathbf{r}^w - \mathbf{r}^l))(\mathcal{L}_{t-1}^\mathbf{x} + \mathcal{L}_{t-1}^\mathbf{v}) + (1 - \sigma(\mathbf{r}^w - \mathbf{r}^l))(\bar{\mathcal{L}}_{t-1}^\mathbf{x} + \bar{\mathcal{L}}_{t-1}^\mathbf{v})\big]$ *is a variational upper bound of:*

$$
\begin{aligned}
\mathcal{L}_{E^2PO}(\theta) = -\mathbb{E}_{(\mathbf{p}, \mathbf{m}^w, \mathbf{m}^l)\sim\mathcal{D}}\Big[&\big(\sigma(\mathbf{r}^w - \mathbf{r}^l)\big)\Big(\log\sigma\big(\beta\log\frac{p_\theta(\mathbf{m}^w|\mathbf{p})}{p_{ref}(\mathbf{m}^w|\mathbf{p})} - \beta\log\frac{p_\theta(\mathbf{m}^l|\mathbf{p})}{p_{ref}(\mathbf{m}^l|\mathbf{p})}\big)\Big) \\
&+ \big(1 - \sigma(\mathbf{r}^w - \mathbf{r}^l)\big)\Big(\log\sigma\big(\beta\log\frac{p_\theta(\mathbf{m}^w|\mathbf{p})}{p_{ref}(\mathbf{m}^w|\mathbf{p})} - \beta\log\frac{p_\theta(\mathbf{m}^l|\mathbf{p})}{p_{ref}(\mathbf{m}^l|\mathbf{p})}\big)\Big)\Big].
\end{aligned}
\tag{13}
$$

We refer readers to Appendix S2 of Diffusion-DPO [Wallace et al., 2023] for the full proof. The bound is derived from Jensen's inequality and the convexity of the function $-\log\sigma$.

**Lemma C.2.** *The objective function Equation* (13) *optimizes the KL-divergence* $\mathbb{D}_{\text{KL}}\big(\hat{p}^*(\mathbf{m}|\mathbf{p})||\hat{p}_\theta(\mathbf{m}|\mathbf{p})\big)$, *where* $\hat{p}^*(\mathbf{m}|\mathbf{p}) \propto p_{\text{ref}}(\mathbf{m}|\mathbf{p})\exp(r(\mathbf{m}, \mathbf{p}))$ *and* $\hat{p}_\theta(\mathbf{m}|\mathbf{p}) \propto p_{\text{ref}}(\mathbf{m}|\mathbf{p})\left(\frac{p_\theta(\mathbf{m}|\mathbf{p})}{p_{\text{ref}}(\mathbf{m}|\mathbf{p})}\right)^\beta$.

*Proof.* First of all, we can rewrite the objective Equation (13) in the following form, expanding the sigmoid function:

$$
\begin{aligned}
\mathcal{L}_{\mathrm{E}^2\mathrm{PO}}(\theta) &= \mathbb{E}_{\mathbf{p}\sim\mathcal{D}}\mathbb{E}_{p_{\mathrm{ref}}(\mathbf{m}_{1:2}|\mathbf{p})}\left[-\sum_{i=1}^{2}\frac{e^{r(\mathbf{p},\mathbf{m}_i)}}{\sum_{j=1}^{2}e^{r(\mathbf{p},\mathbf{m}_j)}}\log\frac{e^{\beta\log\frac{p_\theta(\mathbf{m}_i|\mathbf{p})}{p_{\mathrm{ref}}(\mathbf{m}_i|\mathbf{p})}}}{\sum_{j=1}^{2}e^{\beta\log\frac{p_\theta(\mathbf{m}_j|\mathbf{p})}{p_{\mathrm{ref}}(\mathbf{m}_j|\mathbf{p})}}}\right]\\
&= \mathbb{E}_{\mathbf{p}\sim\mathcal{D}}\mathbb{E}_{p_{\mathrm{ref}}(\mathbf{m}_{1:2}|\mathbf{p})}\left[-\sum_{i=1}^{2}\frac{e^{r(\mathbf{p},\mathbf{m}_i)}}{\sum_{j=1}^{2}e^{r(\mathbf{p},\mathbf{m}_j)}}\log\frac{e^{\log\left(\frac{p_\theta(\mathbf{m}_i|\mathbf{p})}{p_{\mathrm{ref}}(\mathbf{m}_i|\mathbf{p})}\right)^\beta}}{\sum_{j=1}^{2}e^{\log\left(\frac{p_\theta(\mathbf{m}_j|\mathbf{p})}{p_{\mathrm{ref}}(\mathbf{m}_j|\mathbf{p})}\right)^\beta}}\right] \quad (14)\\
&= \mathbb{E}_{\mathbf{p}\sim\mathcal{D}}\mathbb{E}_{p_{\mathrm{ref}}(\mathbf{m}_{1:2}|\mathbf{p})}\left[-\sum_{i=1}^{2}\frac{e^{r(\mathbf{p},\mathbf{m}_i)}}{\sum_{j=1}^{2}e^{r(\mathbf{p},\mathbf{m}_j)}}\log\frac{\left(\frac{p_\theta(\mathbf{m}_i|\mathbf{p})}{p_{\mathrm{ref}}(\mathbf{m}_i|\mathbf{p})}\right)^\beta}{\sum_{j=1}^{2}\left(\frac{p_\theta(\mathbf{m}_j|\mathbf{p})}{p_{\mathrm{ref}}(\mathbf{m}_j|\mathbf{p})}\right)^\beta}\right]
\end{aligned}
$$

By the definition $\hat{p}_\theta(\mathbf{m}|\mathbf{p})\propto p_{\mathrm{ref}}^{1-\beta}(\mathbf{m}|\mathbf{p})p_\theta^\beta(\mathbf{m}|\mathbf{p})$, we have $\frac{\hat{p}_\theta(\mathbf{m}|\mathbf{p})}{p_{\mathrm{ref}}(\mathbf{m}|\mathbf{p})}\propto\left(\frac{p_\theta(\mathbf{m}|\mathbf{p})}{p_{\mathrm{ref}}(\mathbf{m}|\mathbf{p})}\right)^\beta$ (by dividing both sides with $p_{\mathrm{ref}}(\mathbf{m}|\mathbf{p})$). Then we can substitute this equation and rewrite $\mathcal{L}_{\mathrm{E}^2\mathrm{PO}}(\theta)$:

$$
\begin{aligned}
\mathcal{L}_{\mathrm{E}^2\mathrm{PO}}(\theta) &= \mathbb{E}_{\mathbf{p}\sim\mathcal{D}}\mathbb{E}_{p_{\mathrm{ref}}(\mathbf{m}_{1:2}|\mathbf{p})}\left[-\sum_{i=1}^{2}\frac{e^{r(\mathbf{p},\mathbf{m}_i)}}{\sum_{j=1}^{2}e^{r(\mathbf{p},\mathbf{m}_j)}}\log\frac{\left(\frac{p_\theta(\mathbf{m}_i|\mathbf{p})}{p_{\mathrm{ref}}(\mathbf{m}_i|\mathbf{p})}\right)^\beta}{\sum_{j=1}^{2}\left(\frac{p_\theta(\mathbf{m}_j|\mathbf{p})}{p_{\mathrm{ref}}(\mathbf{m}_j|\mathbf{p})}\right)^\beta}\right]\\
&= \mathbb{E}_{\mathbf{p}\sim\mathcal{D}}\mathbb{E}_{p_{\mathrm{ref}}(\mathbf{m}_{1:2}|\mathbf{p})}\left[-\sum_{i=1}^{2}\frac{e^{r(\mathbf{p},\mathbf{m}_i)}}{\sum_{j=1}^{2}e^{r(\mathbf{p},\mathbf{m}_j)}}\log\frac{\frac{\hat{p}_\theta(\mathbf{m}_i|\mathbf{p})}{p_{\mathrm{ref}}(\mathbf{m}_i|\mathbf{p})}}{\sum_{j=1}^{2}\frac{\hat{p}_\theta(\mathbf{m}_j|\mathbf{p})}{p_{\mathrm{ref}}(\mathbf{m}_j|\mathbf{p})}}\right] \quad (15)
\end{aligned}
$$

Since $p_{\mathrm{ref}}(\cdot|\mathbf{p})$ is supervised fine-tuned on samples $\{\mathbf{m}_i\}_{i=1}^2$, we can assume $\{\mathbf{m}_i\}_{i=1}^2$ takes most of the probability mass and thus $\mathbb{E}_{p_{\mathrm{ref}}(\mathbf{m}|\mathbf{p})}\approx\mathbb{E}_{p_{\mathrm{ref}}(\mathbf{m}_{1:2}|\mathbf{p})}$. Then we have the following approximation:

$$
\begin{aligned}
\sum_{j=1}^{2}\frac{\hat{p}_\theta(\mathbf{m}_j|\mathbf{p})}{p_{\mathrm{ref}}(\mathbf{m}_j|\mathbf{p})} &\approx 2\mathbb{E}_{p_{\mathrm{ref}}(\mathbf{m}|\mathbf{p})}\left[\frac{\hat{p}_\theta(\mathbf{m}|\mathbf{p})}{p_{\mathrm{ref}}(\mathbf{m}|\mathbf{p})}\right] = 2\sum_{\mathbf{m}\in\mathcal{M}}p_{\mathrm{ref}}(\mathbf{m}|\mathbf{p})\frac{\hat{p}_\theta(\mathbf{m}|\mathbf{p})}{p_{\mathrm{ref}}(\mathbf{m}|\mathbf{p})} = 2\sum_{\mathbf{m}\in\mathcal{M}}\hat{p}_\theta(\mathbf{m}|\mathbf{p}) = 2,\\
\sum_{j=1}^{2}e^{r(\mathbf{p},\mathbf{m}_j)} &\approx 2\mathbb{E}_{p_{\mathrm{ref}}(\mathbf{m}|\mathbf{p})}\left[e^{r(\mathbf{p},\mathbf{m})}\right] = 2\sum_{\mathbf{m}\in\mathcal{M}}p_{\mathrm{ref}}(\mathbf{m}|\mathbf{p})e^{r(\mathbf{p},\mathbf{m})} = 2Z(\mathbf{p}).
\end{aligned}
$$

Then we can plug the above results into Equation (15) and further simplify $\mathcal{L}_{\mathrm{E}^2\mathrm{PO}}$:

$$
\begin{aligned}
\mathcal{L}_{\mathrm{E}^2\mathrm{PO}}(\theta) &= \mathbb{E}_{\mathbf{p}\sim\mathcal{D}}\mathbb{E}_{p_{\mathrm{ref}}(\mathbf{m}_{1:2}|\mathbf{p})}\left[-\sum_{i=1}^{2}\frac{e^{r(\mathbf{p},\mathbf{m}_i)}}{2Z(\mathbf{p})}\log\frac{\hat{p}_\theta(\mathbf{m}_i|\mathbf{p})}{2p_{\mathrm{ref}}(\mathbf{m}_i|\mathbf{p})}\right]\\
&= \mathbb{E}_{\mathbf{p}\sim\mathcal{D}}\mathbb{E}_{p_{\mathrm{ref}}(\mathbf{m}_{1:2}|\mathbf{p})}\left[-\sum_{i=1}^{2}\frac{e^{r(\mathbf{p},\mathbf{m}_i)}}{2Z(\mathbf{p})}\log\left(\frac{\hat{p}_\theta(\mathbf{m}_i|\mathbf{p})}{p_{\mathrm{ref}}(\mathbf{m}_i|\mathbf{p})\frac{e^{r(\mathbf{p},\mathbf{m}_i)}}{Z(\mathbf{p})}}\frac{e^{r(\mathbf{p},\mathbf{m}_i)}}{2Z(\mathbf{p})}\right)\right]\\
&= \mathbb{E}_{\mathbf{p}\sim\mathcal{D}}\mathbb{E}_{p_{\mathrm{ref}}(\mathbf{m}_{1:2}|\mathbf{p})}\left[-\sum_{i=1}^{2}\frac{e^{r(\mathbf{p},\mathbf{m}_i)}}{2Z(\mathbf{p})}\log\left(\frac{\hat{p}_\theta(\mathbf{m}_i|\mathbf{p})}{p_{\mathrm{ref}}(\mathbf{m}_i|\mathbf{p})\frac{e^{r(\mathbf{p},\mathbf{m}_i)}}{Z(\mathbf{p})}}\right) - \sum_{i=1}^{2}\frac{e^{r(\mathbf{p},\mathbf{m}_i)}}{2Z(\mathbf{p})}\log\left(\frac{e^{r(\mathbf{p},\mathbf{m}_i)}}{2Z(\mathbf{p})}\right)\right],
\end{aligned}
$$

where the second term remains constant $C$ to $\theta$, and thus can be omitted when analyzing the optimization for $\theta$. Notice the normalized form of $\hat{p}^*(\mathbf{m}|\mathbf{p}) = \frac{1}{Z(\mathbf{p})}p_{\mathrm{ref}}(\mathbf{m}|\mathbf{p})e^{r(\mathbf{p},\mathbf{m})}$, we replace

$\frac{1}{Z(\mathbf{p})} p_{\text{ref}}(\mathbf{m}|\mathbf{p}) e^{r(\mathbf{p},\mathbf{m})}$ with $\hat{p}^*$ and further simplify the above equation:

$$
\begin{aligned}
\mathcal{L}_{\text{E}^2\text{PO}}(\theta) &= \mathbb{E}_{\mathbf{p}\sim\mathcal{D}}\left[ -\frac{1}{2}\sum_{i=1}^{2}\left[\frac{e^{r(\mathbf{p},\mathbf{m}_i)}}{Z(\mathbf{p})}\log\frac{\hat{p}_\theta(\mathbf{m}_i|\mathbf{p})}{\hat{p}^*(\mathbf{m}_i|\mathbf{p})}\right] + C \right] \\
&= \mathbb{E}_{\mathbf{p}\sim\mathcal{D}}\left[ -\mathbb{E}_{p_{\text{ref}}(\mathbf{m}|\mathbf{p})}\left[\frac{e^{r(\mathbf{p},\mathbf{m})}}{Z(\mathbf{p})}\log\frac{\hat{p}_\theta(\mathbf{m}|\mathbf{p})}{\hat{p}^*(\mathbf{m}|\mathbf{p})}\right] + C \right] \\
&= \mathbb{E}_{\mathbf{p}\sim\mathcal{D}}\left[ -\sum_{\mathbf{m}\in\mathcal{M}} p_{\text{ref}}(\mathbf{m}|\mathbf{p})\frac{e^{r(\mathbf{p},\mathbf{m})}}{Z(\mathbf{p})}\log\frac{\hat{p}_\theta(\mathbf{m}|\mathbf{p})}{\hat{p}^*(\mathbf{m}|\mathbf{p})} + C \right] \\
&= \mathbb{E}_{\mathbf{p}\sim\mathcal{D}}\left[ -\sum_{\mathbf{m}\in\mathcal{M}} \hat{p}^*(\mathbf{m}|\mathbf{p})\log\frac{\hat{p}_\theta(\mathbf{m}|\mathbf{p})}{\hat{p}^*(\mathbf{m}|\mathbf{p})} + C \right] \\
&= \mathbb{E}_{\mathbf{p}\sim\mathcal{D}}\left[ \mathbb{D}_{\text{KL}}(\hat{p}^*(\cdot|\mathbf{p})\|\hat{p}_\theta(\cdot|\mathbf{p})) + C \right],
\end{aligned}
$$

which completes the proof of Lemma C.2. $\qquad\square$

# D    Implementation Details

**Data.** Following [Guan et al., 2023], proteins and ligands are expressed with atom coordinates and a one-hot vector containing the atom types. For proteins, each atom type is represented by a one-hot vector covering 20 distinct amino acids. Ligand atoms are encoded using a one-hot vector that discriminates among several elements, specifically H, C, N, O, F, P, S, Cl. Additionally, a one-dimensional binary flag is incorporated to differentiate whether atoms are part of the protein or the ligand. We further apply two separate single-layer Multi-Layer Perceptrons (MLPs) to transform the input data into 128-dimensional latent spaces, providing a compact and informative representation for subsequent computational stages.

**Preference Pair Generation.** For each synthetic molecule, we first locate its corresponding protein binding site and compute reward according to user-defined reward function for all synthetic molecules of the corresponding the binding site. We select a losing sample with lower reward and construct the preference. The selection process is detailed in Appendix E.

**Architecture.** We follow the same architecture as IPDiff [Huang et al., 2023], which includes a learnable diffusion denoising model $\phi_{\theta1}$, learnable neural network $\phi_{\theta2}$ and pretrained interaction prior network IPNET. The architecture of all models used in our method is the same as IPDiff.

**Pretraining Details.** Following existing work, we adopted the Adam optimizer with a learning rate of 0.001 and parameters $\beta$ values of (0.95, 0.999). The training was conducted with a batch size of 4 and a gradient norm clipping value of 8. To balance the losses for atom type and atom position, we applied a scaling factor $\lambda$ of 100 to the atom type loss. Additionally, we introduced Gaussian noise with a standard deviation of 0.1 to the protein atom coordinates as a form of data augmentation. Our parameterized diffusion denoising model, IPDiff was trained on a single NVIDIA A6000 GPU and achieved convergence within 200k steps.

**Training Details.** For finetuning, the pre-trained diffusion model is further fine-tuned via the gradient descent method Adam with init learning rate=5e-6, betas=(0.95,0.999). We keep other setting the same as pretraining. We use $\beta = 5$ in Equation (5). We trained our model with one NVIDIA GeForce GTX A100 GPU, and it could converge within 30k steps.

# E    More Experimental Results

**Effect of diffusion steps.** In fig. 6, we present a comprehensive ablation study examining the impact of diffusion steps on the optimization of molecular properties using our novel ALIDIFF framework. The visualizations at the top of the figure showcase the progressive refinement of molecular structures across increasing diffusion steps ($t = 200$ to $t = 1000$). These images clearly illustrate how our model gradually enhances the molecular fitting within the target binding site, which is critical for improving drug efficacy. The plotted data below provides a quantitative analysis of QED, SA, Vina

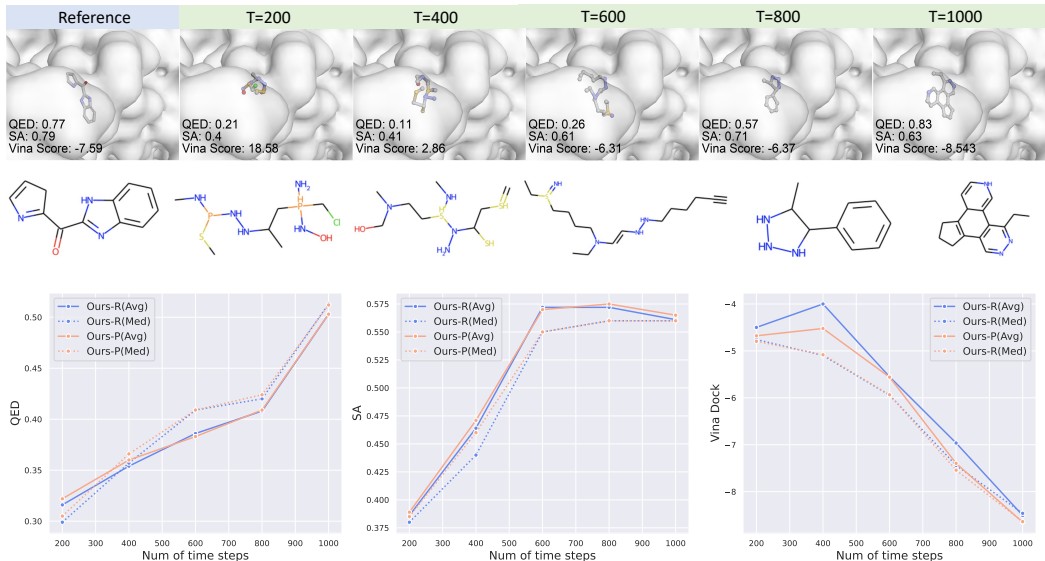

Figure 6: Ablation study on diffusion steps. The top shows a visualization of the generated molecule (4aua) under different time step. The bottom reports QED, SA and Vina Dock are reported under different diffusion steps(200, 400, 600, 800 and 1000). Blue lines represent ALIDIFF-DPO and Red lines represent ALIDIFF-E$^2$PO.

Dock across all test targets. Notably, both ALIDIFF (P) and ALIDIFF (R) demonstrate significant improvements in QED and SA scores as the number of diffusion steps increases and exhibit a notable decrease in Vina Dock. Particularly, ALIDIFF-E$^2$PO model have shown better performance across all three metrics, with significant improvement on binding affinity across the diffusion steps.

**Lipinski.** We further compared Lipinski's Rule of Five [Lipinski et al., 2012] across all comparison methods. Lipinski's Rule of Five is another measure-

Table 5: Lipinski results for all methods.

| Methods | ALIDIFF | IPDiff | TargetDiff | AR | Pocket2Mol | Reference |
|---|---|---|---|---|---|---|
| Avg. Lipinski (↑) | 4.48 | 4.52 | 4.51 | 4.75 | **4.88** | 4.27 |

ment for assessing drug-likeness besides QED, and we would like to incorporate this metric to validate our performance in generating drug-like molecules. The results of Lipinski's scores are reported in Table 5. The results are consistent with our evaluation using QED score, as all diffusion-based models are not achieving high drug-likeness. We maintain similar drug-likeness as our backbone models targetDiff and IPDiff.

Table 6: Ablation study results with different choice of $\mathbf{m}^l$.

| Choice of $\mathbf{m}^l$ | Vina Score (↓) | | Vina Min (↓) | | Vina Dock (↓) | | High Affinity(↑) | | QED(↑) | | SA(↑) | | Diversity(↑) | |
|---|---|---|---|---|---|---|---|---|---|---|---|---|---|---|
| | Avg. | Med. | Avg. | Med. | Avg. | Med. | Avg. | Med. | Avg. | Med. | Avg. | Med. | Avg. | Med. |
| worst | **-7.07** | **-7.95** | **-8.09** | **-8.17** | **-8.90** | -8.81 | **73.4%** | **81.4%** | 0.50 | 0.50 | **0.57** | **0.56** | 0.73 | 0.71 |
| best | -6.80 | -7.66 | -7.83 | -7.69 | -8.64 | -8.05 | 70.2% | 76.8% | 0.50 | **0.52** | 0.56 | 0.55 | **0.74** | 0.71 |
| random | -6.96 | -7.82 | -8.03 | -8.00 | -8.77 | -8.20 | 72.1% | 77.8% | 0.50 | 0.51 | 0.56 | 0.55 | **0.74** | 0.72 |
| median | -6.96 | -7.85 | -8.01 | -7.96 | -8.80 | -8.24 | 72.5% | 78.9% | 0.50 | 0.51 | **0.57** | 0.55 | **0.74** | 0.72 |

**Choice of $\mathbf{m}^l$.** Our generated dataset is obtained by directly transforming a standard labeled dataset into a pairwise preference dataset. Yet the binding affinity labels are continuous values where sometimes the difference between preferred and dispreferred is minimal. Therefore, the effect of energy preference optimization is highly sensitive to the overall data quality. Table 6 compares the performance of applying different strategies for selecting the dispreferred samples. "worst" indicates that the losing sample has the worst score from the user-defined reward function (lowest binding affinity). "best" suggests that the losing sample has the second-to-highest binding affinity(besides the preferred one). "random" and "median" mean that the losing samples are extracted randomly or from the median. Vina Score, Vina Min, Vina Dock, QED, SA, and Diversity are reported as average (Avg.) and median (Med.) values. Overall, the "worst" strategy, selecting the least favorable sample based on optimization objectiveness, consistently achieves the best performance in binding affinity metrics (Vina Score, Vina Min, and Vina Dock), while maintaining competitive drug-likeness (QED)

and synthetic accessibility (SA). The "best" strategy, which may involve selecting the most favorable samples, performs poorly overall, which implies that energy preference optimization works better when there exists a larger discrepancy between $\mathbf{r}^w$ and $\mathbf{r}^l$. This allows the model to learn how to favor to $\mathbf{m}^w$ and avoid $\mathbf{m}^l$ during the finetuning process. The "random" and "median" strategies show intermediate performance, suggesting that a strategic approach to sample selection can significantly impact the efficacy of the resulting models.

# F   More Visualizations

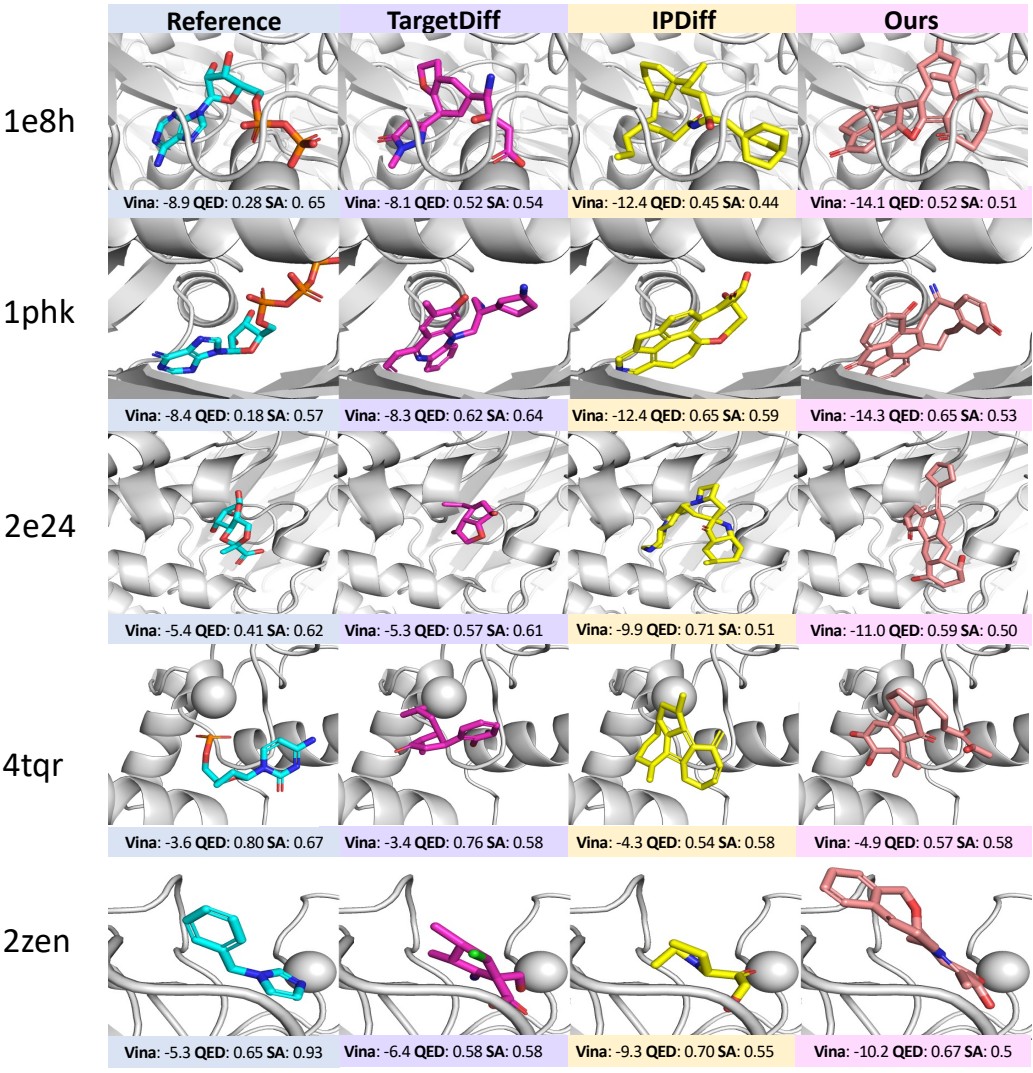

Figure 7: More visualizations of generated ligands for protein pockets generated by TargetDiff, IPDiff, and ALIDIFF.

