# OpenReview forum: "Aligning Target-Aware Molecule Diffusion Models with Exact Energy Optimization"
_NeurIPS.cc/2024/Conference — NeurIPS 2024 poster_

### Official Review · Reviewer_FKJR · 2024-07-01

**Soundness:** 3
**Presentation:** 4
**Contribution:** 3
**Rating:** 7
**Confidence:** 4

**Summary:**

This paper focuses on generating ligands with desired properties, such as high binding affinity, for the protein-conditioned ligand generation task. The authors propose a Preference Optimization (PO)-based fine-tuning method for pre-trained generative models. They extend DPO (Data-driven Preference Optimization) to propose exact energy optimization that directly utilizes reward values in the loss function. Experiments demonstrate that the proposed method can generate ligand molecules with the desired properties.

**Strengths:**

* Comprehensive ablation studies address readers' concerns.
* An efficient objective function is designed for tasks where reward values are directly available, such as binding energy.
* The proposed method's effectiveness is validated using multiple pre-trained models, both IPDiff and TargetDiff, demonstrating its generalizability.
* The investigation into preference pair selection is interesting and shows room for future research.
* The proposed method is well-designed.
* The paper is well-structured and easy to read.

**Weaknesses:**

* The practical impact of the achieved improvement needs to be clearly quantified.
* There is no consideration or results for comparison or combination with methods other than DPO, such as IPO or SLiC.

**Questions:**

* The proposed objective function seems to be heavily influenced by the scale of the reward value r. Was any adjustment necessary? In particular, was simple addition sufficient for the Affinity + SA experiment that combined multiple rewards? Additionally, is it easy to incorporate other rewards, such as diversity?
* The paper states that if $r^w \gg r^l$, then $\sigma(r^w - r^l) \approx 1$. However, wouldn't it be more accurate to say that if $r^w - r^l \ge 1$, then $\sigma(r^w - r^l) \approx 1$? What is the actual distribution of $r^w - r^l$ in the training data?

**Limitations:**

The proposed objective function cannot be used when reward values are not explicitly available, making it unsuitable for cases involving manually binary-labeled results.

---

> ### Author Rebuttal · Authors · 2024-08-07
>
> Thank you for your constructive feedback and questions! The replies to your questions are listed below:
>
> > **[Q1] Practical impact**
>
> Our benchmark results showcase the practical impact of optimizing molecules with user-defined reward functions. A lower binding energy between a protein and a ligand indicates a stronger and more stable interaction between the two molecules. In drug design, binding energy is a critical factor that influences the efficacy of a ligand in binding to its target protein. Our benchmark results suggest that while we maintain similar molecular properties as baseline models, we are able to improve binding affinity by a notable margin.
>
> In addition to the benchmarks presented in the paper, we have explored several practical case studies to demonstrate the effectiveness of the proposed method for generating ligands with desired properties. In Fig 4, we showcased visualizations of reference molecules and generated ligands for protein pockets (1l3l, 2e24). For instance, protein 1l3l is a transcription factor involved in quorum sensing, a process by which bacteria communicate and coordinate behavior based on their population density. Designing drugs targeting the 1l3l protein can prevent the expression of virulence genes. and suppress unwanted bacterial activity. These case studies highlight the method's potential to address real-world challenges in drug design and discovery.  We will incorporate this discussion into final version.
>
> > **[Q2] Comparison with other RLHF methods**
>
> Thank you for your insightful comment regarding exploring more RLHF methods. We appreciate your suggestion and added additional experiments incorporating the IPO method [1]. Below, we provide an updated comparison table that includes results for the DPO, E²PO, and IPO:
>
> | Method   | Vina Score (↓) | Vina Min (↓) | Vina Dock (↓) | High Affinity (↑) | QED (↑) | SA (↑) | Diversity (↑) |
> |-|-|-|-|-|-|-|-|
> | **AliDiff-DPO** Avg.  | -6.81 | -7.75  | -8.58  | 69.7%  | **0.50**| 0.56  | **0.74**  |
> |  Med.  | -7.62  | -7.79  |  -8.55  | 71.1% | **0.51**| **0.57** | **0.72**     |
> | **AliDiff-E²PO** Avg.  | **-7.07**  | **-8.09** |  **-8.90**  |  **73.4%** | **0.50**    | **0.57** | 0.73  |
> |   Med. | **-7.95**  |  **-8.17** |  **-8.81** |  **81.4%** | 0.50 | 0.56  | 0.71|
> | **AliDiff-IPO** Avg.  | -6.93 |  -7.80 | -8.68 |  71.7% | **0.50**| **0.57**  | 0.73  |
> |  Med.  | -7.82  | -7.92  | -8.62  | 78.9%| **0.51**| 0.56  | **0.72** |
>
> AliDiff-E²PO remains the top performer in terms of binding affinity and high-affinity metrics, which further supports the effectiveness of incorporating exact preference optimization. We will add this study to the final version.
>
> > **[W1] Scale of the reward value r**
>
> 1. Yes, the proposed objective function is influenced by the scale of the reward value r. Therefore, we conducted a sensitivity analysis to determine the ideal r, as shown in Fig. 5. When combining multiple reward objectives Vina and SA, we applied a simple weighted sum approach. Since the numerical scale of binding affinity (Vina score) and molecular property (SA) are different, we evaluated different weighing factors and the results are shown as follows:
>
> | Weight |Avg. Vina Score (↓) |Avg. Vina Min (↓) | Avg.Vina Dock (↓) | Avg. High Affinity (↑) | Avg. QED (↑) | Avg. SA (↑) | Avg. Diversity (↑) |
> |-|-|-|-|-|-|-|-|
> | 1   | **-6.99** |  **-8.02**   | **-8.89**  | **73.3%** | 0.50  | 0.57  | 0.73  |
> | 10   | -6.87  | -8.00   | -8.81 | 72.7% | 0.52|**0.60**| **0.74** |
> | 100   |  -6.78 | -7.90   | -8.71  | 71.7% | **0.53**|**0.60**| 0.73 |
>
> We leave finding the optimal approach for multi-objective optimization as a promising future direction, as it will take advantage of multiple objectives and allow for better optimization from each objective.
>
> 2. Yes, it is indeed possible to align the pre-trained diffusion model with other reward functions. The results are shown below, where we combine affinity with QED and SA. By incorporating metrics like QED and SA into the reward system, the model generates molecules that balance strong binding affinities with enhanced drug-likeness and synthetic accessibility. Aligning molecules with both binding affinity and QED will improve both metrics, although the improvement in QED is relatively marginal. Therefore, by fine-tuning the pre-trained diffusion model using a multi-objective reward function, we can enhance binding efficacy while preserving their molecular properties. This approach allows for more robust and practical molecule generation in drug discovery applications.
>
> | Choice of Reward | Vina Score (↓) | Vina Min (↓) | Vina Dock (↓) | High Affinity (↑) | QED (↑) | SA (↑) | Diversity (↑) |
> |-|-|-|-|-|-|-|-|
> | Affinity Avg.   | -7.07   | **-8.09**   | -8.90    | 73.4%  | 0.50    | 0.57  | 0.73    |
> | Med.   | -7.95  | **-8.17**   | **-8.81**    | 81.4%       | 0.50    | 0.56  | 0.71 |
> | Affinity + SA Avg. | -6.87 | -8.00   |  -8.81  | 72.7%  | **0.52**|**0.60**| **0.74**  |
> | Med.  | -7.76  |  -8.08   | -8.72    | 80.8%  | **0.55**|**0.59**| **0.73**  |
> | Affinity + QED  Avg.   | **-7.11**  |  -8.02   | **-8.91** | **73.7%** | 0.51|0.57| 0.73  |
> |Med. | **-7.99** | **-8.17**   | -8.72    | 82.0%  | 0.52|0.57| **0.73**  |
>
> > **[W2] Is it be more accurate to say that if $r^w - r^l \ge 1$, then $\sigma(r^w - r^l) \approx 1$? What’s the distribution of  $r^w - r^l$**
>
> Mathematically, for the sigmoid function, we have that $\sigma(1) = 0.7311$. Therefore we would keep the statement that “$\sigma(r^w - r^l) \approx 1$ when $r^w \gg r^l$”. Please let us know if we misunderstand your point.
>
> For the distribution of $r^w - r^l$, the statistics are listed: mean=3.35, median=2.37 and standard deviation=6.22, with boundary values max=120.35 and min=0.00. We will provide a figure for the distribution in our final version.
>
> ------
>
> We hope our response could address your questions!
>
> [1]A general theoretical paradigm to understand learning from human preferences. AISTATS, 2024

---

> > ### Comment · Reviewer_FKJR · 2024-08-11
> > **Thanks**
> >
> > Thank you for the detailed response. It has addressed my concerns.
> > I still believe that this work is worth presenting.
> >
> > > Mathematically, for the sigmoid function, ...
> >
> > Sorry, it is my misunderstanding. You are correct.

---

> > > ### Author Response · Authors · 2024-08-13
> > >
> > > Thank you very much for your timely reply and recognition of our efforts! We appreciate your valuable suggestions and will incorporate the discussions and results to the final version. Please let me know if you have any other questions!
> > >
> > > Sincerely, Authors

---

### Official Review · Reviewer_bNcB · 2024-07-08

**Soundness:** 3
**Presentation:** 2
**Contribution:** 3
**Rating:** 5
**Confidence:** 2

**Summary:**

This paper presents ALIDIFF, a framework that aligns pre-trained target-aware molecule diffusion models with desired functional properties using preference optimization. The key contribution of ALIDIFF is the Exact Energy Preference Optimization method, which precisely aligns diffusion models to regions with lower binding energy and better structural rationality based on user-defined reward functions. Extensive experiments on the CrossDocked2020 benchmark showcase ALIDIFF's strong performance, generating molecules with state-of-the-art binding energies and maintaining competitive molecular properties. By incorporating user-defined reward functions and improving the Exact Energy Preference Optimization method, ALIDIFF achieves significant advancements in binding affinity.

**Strengths:**

* ALIDIFF utilizes a preference optimization approach to effectively steer the model towards generating chemically and structurally relevant molecules.
* It Demonstrates the effectiveness and robustness of the method through extensive experiments, showing superior binding energies while maintaining strong molecular properties.
* The paper employs AutoDock Vina for accurate binding energy evaluations, ensuring practical outcomes in drug discovery applications.

**Weaknesses:**

* The method leverages IPDiff and further optimizes it using RL, which makes performance improvements easier. Consequently, the methodological novelty appears marginal.
* The paper lacks specific information on the time required for pre-training, fine-tuning, and binding energy computations via AutoDock Vina. This absence makes it difficult to fully assess the advantages and limitations of the proposed method.
  - AutoDock Vina is known to be time-consuming, and RL also requires significant computational time. Including these time details would help illustrate both the strengths and constraints of the proposed approach.

**Questions:**

* Train/test split is 65K/100. It seems that the test set is too small compared to the train set. Is there any reason for that?
* How much time does it take to pre-train, fine-tune and binding energy computations via AutoDock Vina?

**Limitations:**

Yes

---

> ### Author Rebuttal · Authors · 2024-08-07
>
> Thank you for your constructive feedback and questions! The replies to your questions are listed below:
>
> > **[W1] Methodological novelty**
>
> We argue our novelty for not only model design but also method formulation:
> 1. The DPO framework is originally proposed for optimizing language models. In this paper, we propose to optimize the diffusion model with user-preference data to further facilitate drug design with practical pharmaceutical needs, which is already a new perspective in this field.
> 2. Furthermore, we solve key technical challenges for designing optimization frameworks and further introduce exact energy preference optimization, where we further integrate numerical values of reward instead of directly taking a pair of preferred and dispreferred data.
> We believe AliDiff offers considerable contributions to the community.
>
> > **[W2+Q2] Training time**
>
> Our proposed method is not time-consuming. First we want to highlight that training time is reported in the Appendix. The advantage of Ali-Diff is that it can take any pre-trained diffusion model and finetune with preference data. The finetuning process takes 30000 steps with approximately 1916  minutes. During training, we do not need to compute AutoDock Vina because it’s pre-computed in the training dataset. During evaluation, we sampled 100 ligands for each protein, where the evaluation takes 1 hour for each target protein.  We do notice that binding energy computations via AutoDock Vina are time-consuming, and we can also choose to compute binding energy with QVina [1], which will be more time-efficient yet less accurate. All metrics calculate the binding energy between protein and ligands and we can construct alternative datasets using other metrics that are not time-consuming. We will add this discussion to the final version.
>
>
> > **[Q1] Training test split**
>
> We would like to point out that the train/test split we adopt is the benchmark setting from targetDiff and we want to keep this consistent with all other comparison methods. We do notice that 100 test data could be relatively too small, therefore, we also adopt another train/test split from fragment-based methods FLAG[2] and DrugGPS[3] and verify our performance in another setting.
>
> | Methods  | Vina  (↓)  | High Affinity (↑)  | QED (↑)   | SA (↑) | Lip  (↑) |
> |----------|-------|-------|-------|-----------|-------|
> | targetDiff  | -6.92 | 48.2% | 0.47 | 0.58 |  4.42 |
> | IP-Diff  | -7.33 | 64.7% | **0.51** | **0.60** |  **4.48** |
> | AliDiff  | **-7.87** | **68.0%** | 0.50 | 0.56 |  4.43 |
>
> We observe that our proposed method consistently outperforms baseline methods on binding energy-related metrics by a notable margin, which is consistent with our previous conclusion. The inferior performance over drug likeness and synthetic accessibility is mainly due to the nature of atom-based diffusion models, as our baseline models (targetDiff, IP-Diff) also achieve similar performance. We leave combining fragment-based methods with our preference optimization framework as a promising future direction, as it will take advantage of high binding affinity thanks to optimization, while maintaining QED and SA with prior knowledge of fragments.
>
> > **[Q2] Training time for each step.**
>
> We provide our response together with response to weakness2 above.
>
> ------
>
> We hope our response could address your questions!
>
>
>
> [1] Fast, Accurate, and Reliable Molecular Docking with QuickVina 2. Bioinformatics (2015) 31 (13): 2214-2216
>
> [2]: Learning subpocket prototypes for generalizable structure-based drug design. International Conference on Machine Learning. PMLR, 2023.
>
> [3]: Molecule generation for target protein binding with structural motifs. The Eleventh International Conference on Learning Representations. 2023

---

> > ### Comment · Reviewer_bNcB · 2024-08-07
> > **Thanks for the clarification**
> >
> > Thank you for the detailed rebuttal. I have read it thoroughly. Based on the clarified points, I believe that while the methodological novelty may be limited, the practical value is significant. Therefore, I will raise my score to 5. Thank you for the excellent research.

---

> > > ### Author Response · Authors · 2024-08-09
> > >
> > > Dear reviewer bNcB,
> > >
> > > Thank you very much for your timely reply and recognition of our efforts! We appreciate your valuable suggestions and will incorporate the discussions and results to the final version. Please let me know if you have any other questions!
> > >
> > > Sincerely,
> > > Authors

---

### Official Review · Reviewer_fWJ4 · 2024-07-12

**Soundness:** 3
**Presentation:** 4
**Contribution:** 3
**Rating:** 6
**Confidence:** 4

**Summary:**

This paper proposes a novel and general alignment framework to align pretrained target diffusion models with preferred functional properties, named AliDiff. AliDiff adjusts the target-conditioned chemical distribution toward regions characterized by lower binding energy and structural rationality. AliDiff can generate molecules with lower binding energies.

**Strengths:**

- Structure-based drug design is an important scientific problem. In practice, it is more important to control the generation process, instead of learning the distribution of all drug candidates.
- The authors propose a novel preference optimization framework (E2PO) to fine-tune the pre-trained diffusion model with RL.
- The authors address the over-optimization issue with regularization on preference maximization, and propose an analytical guarantee.
- AliDiff achieves strong performance on the CrossDocked2020 benchmark.

**Weaknesses:**

- The experiments are only conducted on the CrossDocked2020 dataset. Are there any alternative datasets?
- In addition to well-established benchmarks, is it possible to show several practical case studies to verify the effectiveness of the proposed algorithms. For example, generating ligands for the target of SARS-CoV-2.

**Questions:**

See weaknesses

---

> ### Author Rebuttal · Authors · 2024-08-07
>
> Thank you for your constructive feedback and questions! The replies to your questions are listed below:
>
> > **[W1] Alternative dataset**
>
> AliDiff’s training requires the dataset to have multiple ligands with one single protein. CrossDocked2020 is one of the largest synthetic datasets where it has multiple liagnds bind to the same protein. We also follow your suggestion and test AliDiff on another high-quality protein-ligand pair dataset Binding MOAD [1]. We train the models by adopting DiffSBDD as the baseline and follow the same implementation details as our previous setting. The results are as follows:
>
> | Metrics         | GraphBP  | DiffSBDD | AliDiff-D|
> |---------|--------|----------|------------------|
> | Vina      | -4.84   | -7.31          | -7.62 |
> | QED             | 0.51    | 0.54            | 0.53 |
> | SA              | 0.31    | 0.62            | 0.60 |
> | Lipinski        | 4.95    | 4.78            | 4.72 |
> | Diversity       | 0.83    | 0.74            | 0.71 |
>
> We observe that our proposed method consistently outperforms baseline methods on binding energy-related metrics, and slightly inferior performance over drug likeness and synthetic accessibility compared with our backbone model DiffSBDD. This is consistent with our previous conclusion on CrossDocked2020. We significantly improved all binding affinity related metrics without significant sacrifice on molecular properties. We will add results on this alternative dataset to the final version.
>
>
>
> > **[W2] Pratical case studies**
>
> In addition to the benchmarks presented in the paper, we have explored several practical case studies to further demonstrate the effectiveness of the proposed method for generating ligands with desired properties. In Fig 4, we showcased visualizations of reference molecules and generated ligands for protein pockets (1l3l, 2e24) Protein 1L3L[2] is a transcription factor involved in quorum sensing, a process by which bacteria communicate and coordinate behavior based on their population density. Designing drugs targeting the 1L3Lprotein can prevent the expression of virulence genes. and suppress unwanted bacterial activity. Protein 2E24 [3] specifically targets and cleaves the pyruvated side chains of xanthan, a complex bacterial heteropolysaccharide.  Drug design targeting this enzyme could potentially enhance industrial applications of xanthan by enabling more precise modifications of its structure, thereby improving the rheological properties of xanthan-based products. These case studies highlight the method's potential to address real-world challenges in drug design and discovery.
> Regarding generating ligands for SARS-CoV-2, our proposed method will first preprocess the protein with a given pocket to extract the binding site. Since SARS-CoV-2 is a relatively new target and few empirical studies have been conducted, we plan to leave SARS-CoV-2 as future test case and incorporate more practical scenarios as our future directions.
>
> ------
>
> We hope our response could address your questions!
>
>
> [1] Binding MOAD, a high-quality protein–ligand database. Nucleic acids research 36.suppl_1 (2007): D674-D678.
>
> [2]: Structure of a bacterial quorum-sensing transcription factor complexed with pheromone and DNA. Nature 417.6892 (2002): 971-974.
>
> [3]: A structural factor responsible for substrate recognition by Bacillus sp. GL1 xanthan lyase that acts specifically on pyruvated side chains of xanthan. Biochemistry 46.3 (2007): 781-791.

---

> > ### Comment · Reviewer_fWJ4 · 2024-08-08
> >
> > I'm glad to see that the author's response has addressed my concerns and I maintain the score, which tends to acceptance.

---

> > > ### Author Response · Authors · 2024-08-09
> > >
> > > Dear reviewer fWJ4,
> > >
> > > Thank you very much for your timely reply and recognition of our efforts! We appreciate your valuable suggestions and will incorporate the discussions and results to the final version. Please let me know if you have any other questions!
> > >
> > > Sincerely,
> > > Authors

---

### Official Review · Reviewer_5bxb · 2024-07-13

**Soundness:** 2
**Presentation:** 2
**Contribution:** 3
**Rating:** 4
**Confidence:** 3

**Summary:**

This paper proposes a novel alignment framework, known as ALIDIFF, to align pretrained target diffusion model with preferred functional properties for structure-based drug design. ALIDIFF shifts the target-conditioned chemical distribution towards regions with higher binding affinity and structural rationality, specified by user-defined reward functions, via the preference optimization approach. The experiments show that this method can generate ligands with better binding energy and maintain competitive molecular properties.

**Strengths:**

- This method is novel in using energy preference optimization framework to align molecule generative models with desirable properties in order to generating molecules with high binding affinity to binding targets.
- They analyze the overfitting issue in the preference optimization objective, and propose an
improved exact energy optimization method to yield an exact alignment towards target distribution shifted by reward functions.

**Weaknesses:**

- Lipinski metric should also be reported in Table 1. This is a commonly used metric for measure drug-likeness in previous work.
- Fragment-based methods, such as FLAG and DrugGPS, is missing in Table 1. For comprehensive comparison, I recommend add those two baselines.
- Generated molecules have inferior drug-likeness properties. Isn't it possible to also align the pre-trained diffusion model with high drug-likeness region?

**Questions:**

Please refer to the weakness part.

**Limitations:**

Limitation is discussed.

---

> ### Author Rebuttal · Authors · 2024-08-07
>
> Thank you for your constructive feedback and questions! The replies to your questions are listed below:
>
> > **[W1] Lipinski metric should also be reported in Table 1.**
>
> Thank you for your suggestions. To evaluate drug-likeness, we have compared QED across all comparison methods. Lipinski's Rule of Five[1] is another measurement for assessing drug-likeness, and we would like to incorporate this metric to validate our performance in generating drug-like molecules. The results of the Lipinski's scores are as follows:
>
> | Reference | AR | Pocket2Mol | targetDiff | IP-Diff | Ali-Diff |
> | - | - | - | - | - | - |
> | 4.27 | 4.75 | 4.88 | 4.51 | 4.52 | 4.48 |
>
> The results are consistent with our evaluation using QED score, as all diffusion-based models are not achieving high drug-likeness. We maintain similar drug likeness as our backbone models targetDiff and IP-Diff. We will incorporate the results into our final version.
>
> > **[W2] Fragment-based methods.**
>
> Thank you for pointing out the related works that are fragment-based. We appreciate your feedback and will incorporate DrugGPS[2] and FLAG[3] into our related work section in final version. Regarding the experiment, we did not compare our approach with fragment-based methods because they are not directly comparable since they rely on heavy prior knowledge of fragments, and also utilize a different experimental setting than the benchmark proposed by targetDiff. We have tested our performance using their train/test split, and the results are as follows:
>
> | Methods  | Vina  (↓)  | High Affinity (↑)  | QED (↑)   | SA (↑) | Lip  (↑) |
> |----------|-------|-------|-------|-----------|-------|
> | FLAG 	| -6.96 | 44.5% |  0.55 	| 0.74 | 4.90 |
> | DrugGPS  | -7.28 | 56.5% | **0.61** | **0.74** |  **4.92** |
> | targetDiff  | -6.92 | 48.2% | 0.47 | 0.58 |  4.42 |
> | IP-Diff  | -7.33 | 64.7% | 0.51 | 0.60 |  4.48 |
> | AliDiff  | **-7.87** | **68.0%** | 0.50 | 0.56 |  4.43 |
>
> We observe that our proposed method consistently outperforms fragment-based methods on binding energy-related metrics, which is consistent with our previous conclusion. The inferior performance over drug likeness and synthetic accessibility is mainly due to the nature of atom-based diffusion models, as our baseline model (targetDiff, IP-Diff) also achieves similar performance. We leave combining fragment-based methods with our preference optimization framework as a promising future direction, as it will take advantage of high binding affinity thanks to optimization, while maintaining QED and SA with prior knowledge of fragments.
>
> > **[W3] Align the pre-trained diffusion model with high drug-likeness region.**
>
> Yes, it is indeed possible to align the pre-trained diffusion model with regions of high drug-likeness. We conduct additional experiments and report the results of combining multiple reward objectives as follows:
>
> | Choice of Reward | Vina Score (↓) | Vina Min (↓) | Vina Dock (↓) | High Affinity (↑) | QED (↑) | SA (↑) | Diversity (↑) |
> |-|-|-|-|-|-|-|-|
> | Affinity Avg.   | -7.07   | **-8.09**   | -8.90    | 73.4%  | 0.50    | 0.57  | 0.73    |
> | Med.   | -7.95  | **-8.17**   | **-8.81**    | 81.4%       | 0.50    | 0.56  | 0.71 |
> | Affinity + SA Avg. | -6.87 | -8.00   |  -8.81  | 72.7%  | **0.52**|**0.60**| **0.74**  |
> | Med.  | -7.76  |  -8.08   | -8.72    | 80.8%  | **0.55**|**0.59**| **0.73**  |
> | Affinity + QED  Avg.   | **-7.11**  |  -8.02   | **-8.91** | **73.7%** | 0.51|0.57| 0.73  |
> |Med. | **-7.99** | **-8.17**   | -8.72    | 82.0%  | 0.52|0.57| **0.73**  |
>
>
> By incorporating metrics like QED and SA into the reward system, the model generates molecules that balance strong binding affinities with enhanced drug-likeness and synthetic accessibility. Aligning molecules with both binding affinity and QED will improve both QED and affinity, although the improvement in QED is relatively marginal. Therefore, by fine-tuning the pre-trained diffusion model using a multi-objective reward function, we can enhance the drug-like properties of generated molecules while preserving their binding efficacy. This approach allows for more robust and practical molecule generation in drug discovery applications. We will incorporate the results into our final version.
>
> ------
>
> We hope our response could address your questions!
>
> [1]: Experimental and computational approaches to estimate solubility and permeability in drug discovery and development settings. Advanced drug delivery reviews,64:4–17, 2012.
>
> [2]: Learning subpocket prototypes for generalizable structure-based drug design. International Conference on Machine Learning. PMLR, 2023.
>
> [3]: Molecule generation for target protein binding with structural motifs. The Eleventh International Conference on Learning Representations. 2023

---

> > ### Author Response · Authors · 2024-08-13
> > **Looking forward to feedback during the reviewer-author discussion period**
> >
> > Dear reviewer 5bxb,
> >
> >
> > We would like to first express our sincere gratitude for your time and effort in reviewing our paper. We appreciate your valuable suggestions and will incorporate the discussions and results to the final version.
> >
> >
> > Considering the author-reviewer discussion is ending very soon, could you kindly check our responses and let us know if you have further concerns? We are more than willing to address any other concerns or questions.
> >
> >
> > We would greatly appreciate it if the reviewer would consider adjusting the score, on the basis of our response and other review comments.
> >
> >
> > Thanks again for your constructive reviews!
> >
> >
> > Sincerely,
> > Authors

---

> > ### Author Response · Authors · 2024-08-14
> > **Looking forward to feedback during the reviewer-author discussion period**
> >
> > Thanks again for your time spent reviewing our work and the constructive comments and suggestions. We provided evaluations of two related works and our results showcase that the proposed method consistently outpeforms all baselines on binding affinity metrics, which have also been acknowledged by other reviewers. We also added lipsinki as another measure of drug-likeness and conducted new experiments on aligning molecules with drug-likeness propoerties, as suggested.
> >
> > As the discussion period draws to a close, we would be grateful for your further feedback on our clarifications and recognition of our research's contributions. Thank you again for your valuable suggestions!
> >
> > Best,
> >
> > Authors

---

### Decision · Program_Chairs · 2024-09-25

**Decision:**

Accept (poster)

**Comment:**

This paper contributes a novel diffusion-based model, focusing on generating ligand molecules with high affinity to their protein targets. To this end, Authors proposed a improved Exact Energy Preference Optimization method based on reinforcement raining.

The reviewers have pointed out that the proposed approach has limited computational novelty, but still is of high practical utility. The rebuttal also discussed that the optimization of the affinity is associated with lower drug likeliness scores, which however seems a property of the diffusion models, and was traded for higher drug likeliness by fragment-based methods.

The paper seems of interest to the NeurIPS community.